# Recipe for a General, Powerful, Scalable Graph Transformer

**Ladislav Rampášek**[*]
Mila, Université de Montréal

**Mikhail Galkin**
Mila, McGill University

**Vijay Prakash Dwivedi**
Nanyang Technological University, Singapore

**Anh Tuan Luu**
Nanyang Technological University, Singapore

**Guy Wolf**
Mila, Université de Montréal

**Dominique Beaini**
Valence Discovery, Mila, Université de Montréal

## Abstract

We propose a recipe on how to build a general, powerful, scalable (GPS) graph Transformer with linear complexity and state-of-the-art results on a diverse set of benchmarks. Graph Transformers (GTs) have gained popularity in the field of graph representation learning with a variety of recent publications but they lack a common foundation about what constitutes a good positional or structural encoding, and what differentiates them. In this paper, we summarize the different types of encodings with a clearer definition and categorize them as being *local*, *global* or *relative*. The prior GTs are constrained to small graphs with a few hundred nodes, here we propose the first architecture with a complexity linear in the number of nodes and edges $O(N + E)$ by decoupling the local real-edge aggregation from the fully-connected Transformer. We argue that this decoupling does not negatively affect the expressivity, with our architecture being a universal function approximator on graphs. Our GPS recipe consists of choosing 3 main ingredients: (i) positional/structural encoding, (ii) local message-passing mechanism, and (iii) global attention mechanism. We provide a modular framework GRAPHGPS[1] that supports multiple types of encodings and that provides efficiency and scalability both in small and large graphs. We test our architecture on 16 benchmarks and show highly competitive results in all of them, show-casing the empirical benefits gained by the modularity and the combination of different strategies.

## 1   Introduction

Graph Transformers (GTs) alleviate fundamental limitations pertaining to the sparse message passing mechanism, e.g., over-smoothing [47], over-squashing [1], and expressiveness bounds [61, 45], by allowing nodes to attend to all other nodes in a graph (*global attention*). This benefits several real-world applications, such as modeling chemical interactions beyond the covalent bonds [63], or graph-based robotic control [37]. Global attention, however, requires nodes to be better identifiable within the graph and its substructures [14]. This has led to a flurry of recently proposed fully-connected graph transformer models [14, 36, 63, 44, 31] as well as various positional encoding schemes leveraging spectral features [14, 36, 39] and graph features [16, 9]. Furthermore, standard

---

[*]To whom correspondence should be addressed: `ladislav.rampasek@mila.quebec`
[1]The source code of GRAPHGPS is available at: `https://github.com/rampasek/GraphGPS`.

global attention incurs quadratic computational costs $O(N^2)$ for a graph with $N$ nodes and $E$ edges, that limits GTs to small graphs with up to a few hundred nodes.

Whereas various GT models focus on particular node identifiability aspects, a principled framework for designing GTs is still missing. In this work, we address this gap and propose a recipe for building general, powerful, and scalable (GPS) graph Transformers. The recipe defines (i) embedding modules responsible for aggregating *positional encodings* (PE) and *structural encodings* (SE) with the node, edge, and graph level input features; (ii) processing modules that employ a combination of local message passing and global attention layers (see Figure 1).

The embedding modules organize multiple proposed PE and SE schemes into *local* and *global* levels serving as additional node features whereas positional and structural *relative* features contribute to edge features. The processing modules define a computational graph that allows to balance between message-passing graph neural networks (MPNNs) and Transformer-like global attention, including attention mechanisms *linear* in the number of nodes $O(N)$.

To the best of our knowledge, application of efficient attention models has not yet been thoroughly studied in the graph domain, e.g., only one work [11] explores the adaptation of Performer-style [12] attention approximation on small graphs. Particular challenges emerge with explicit edge features that are incorporated as attention bias in fully-connected graph transformers [36, 63]. Linear transformers do not materialize the attention matrix directly, hence incorporating edge features becomes a non-trivial task. In this work, we hypothesize that explicit edge features are not necessary for the *global graph attention* and adopt Performer [12] and BigBird [66] as exemplary linear attention mechanisms.

Our contributions are as follows. (i) Provide a general, powerful, scalable (GPS) GT blueprint that incorporates positional and structural encodings with local message passing and global attention, visualized in Figure 1. (ii) Provide a better definition of PEs and SEs and organize them into *local, global*, and *relative* categories. (iii) Show that GPS with linear global attention, e.g., provided by Performer [12] or BigBird [66], scales to graphs with several thousand nodes and demonstrates competitive results even without explicit edge features within the attention module, whereas existing fully-connected GTs [36, 63] are limited to graphs of up to few hundred nodes. (iv) Conduct an extensive ablation study that evaluates contribution of PEs, local MPNN, and global attention components in perspective of several benchmarking datasets. (v) Finally, following the success of GraphGym [65] we implement the blueprint within a modular and performant GRAPHGPS package.

## 2 Related Work

**Graph Transformers (GT).** Considering the great successes of Transformers in natural language processing (NLP) [55, 32] and recently also in computer vision [18, 25, 24], it is natural to study their applicability in the graph domain as well. Particularly, they are expected to help alleviate the problems of over-smoothing and over-squashing [1, 54] in MPNNs, which are analogous to the vanishing gradients and lack of long-term dependencies in NLP. Fully-connected Graph Transformer [14] was first introduced together with rudimentary utilisation of eigenvectors of the graph Laplacian as the node positional encoding (PE), to provide the otherwise graph-unaware Transformer a sense of nodes' location in the input graph. Building on top of this work, SAN [36] implemented an invariant aggregation of Laplacian's eigenvectors for the PE, alongside conditional attention for real and virtual edges of a graph, which jointly yielded significant improvements. Concurrently, Graphormer [63, 51] proposed using pair-wise graph distances (or 3D distances) to define relative positional encodings, with outstanding success on large molecular benchmarks. Further, GraphiT [44] used relative PE derived from diffusion kernels to modulate the attention between nodes. Finally, GraphTrans [31] proposed the first hybrid architecture, first using a stack of MPNN layers, before fully-connecting the graph. Since, the field has continued to propose alternative GTs: SAT [9], EGT [29], GRPE [48].

**Positional and structural encodings.** There have been many recent works on PE and SE, notably on Laplacian PE [14, 36, 3, 57, 39], shortest-path-distance [38, 63], node degree centrality [63], kernel distance [44], random-walk SE [16], structure-aware [9, 6, 5], and more. Some works also propose dedicated networks to learn the PE/SE from an initial encoding [36, 16, 39, 9]. To better understand the different PE/SE and the contribution of each work, we categorize them in Table 1 and examine their effect in Section 3.2. In most cases, PE/SE are used as soft bias, meaning they are simply provided as input features. But in other cases, they can be used to direct the messages [3] or create *bridges* between distant nodes [35, 54].

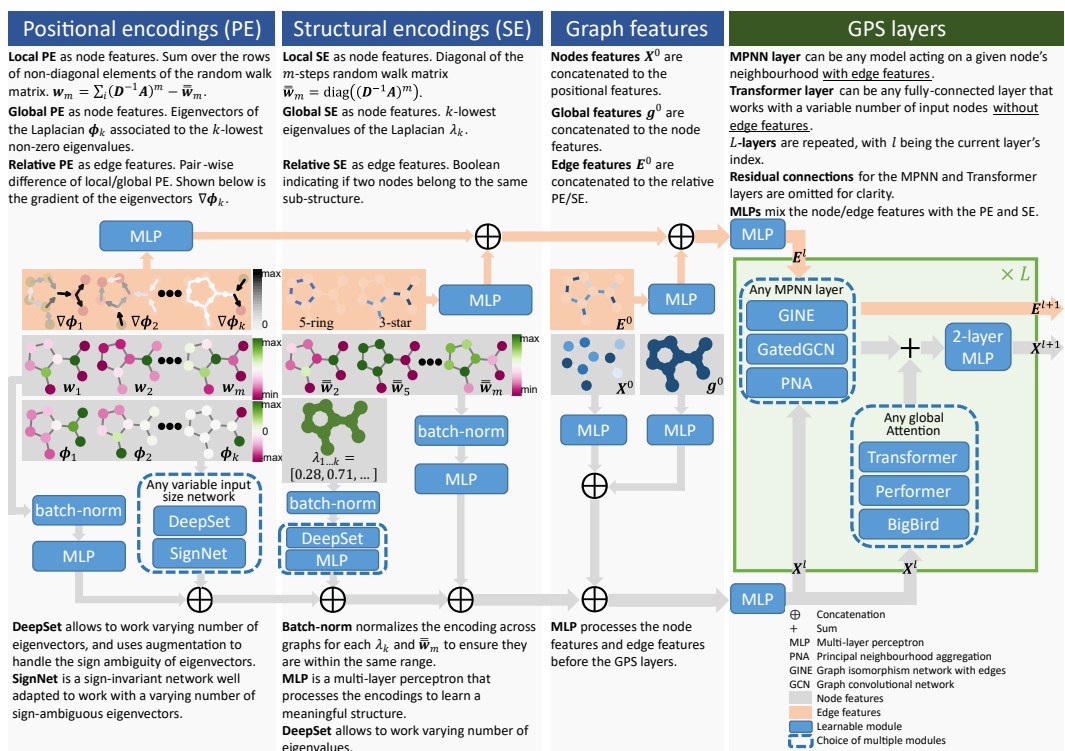

**Positional encodings (PE)**

**Local PE** as node features. Sum over the rows of non-diagonal elements of the random walk matrix. $w_m = \sum_i (D^{-1}A)^m - \bar{w}_m$.

**Global PE** as node features. Eigenvectors of the Laplacian $\phi_k$ associated to the $k$-lowest non-zero eigenvalues.

**Relative PE** as edge features. Pair-wise difference of local/global PE. Shown below is the gradient of the eigenvectors $\nabla\phi_k$.

**DeepSet** allows to work varying number of eigenvectors, and uses augmentation to handle the sign ambiguity of eigenvectors.
**SignNet** is a sign-invariant network well adapted to work with a varying number of sign-ambiguous eigenvectors.

**Structural encodings (SE)**

**Local SE** as node features. Diagonal of the $m$-steps random walk matrix $\bar{w}_m = \text{diag}((D^{-1}A)^m)$.

**Global SE** as node features. $k$-lowest eigenvalues of the Laplacian $\lambda_k$.

**Relative SE** as edge features. Boolean indicating if two nodes belong to the same sub-structure.

**Batch-norm** normalizes the encoding across graphs for each $\lambda_k$ and $\bar{w}_m$ to ensure they are within the same range.
**MLP** is a multi-layer perceptron that processes the encodings to learn a meaningful structure.
**DeepSet** allows to work varying number of eigenvalues.

**Graph features**

**Nodes features** $X^0$ are concatenated to the positional features.

**Global features** $g^0$ are concatenated to the node features.

**Edge features** $E^0$ are concatenated to the relative PE/SE.

**MLP** processes the node features and edge features before the GPS layers.

**GPS layers**

**MPNN layer** can be any model acting on a given node's neighbourhood with edge features.
**Transformer layer** can be any fully-connected layer that works with a variable number of input nodes without edge features.
$L$-**layers** are repeated, with $l$ being the current layer's index.
**Residual connections** for the MPNN and Transformer layers are omitted for clarity.
**MLPs** mix the node/edge features with the PE and SE.

Figure 1: Modular GPS graph Transformer, with examples of PE and SE. Task specific layers for node/graph/edge-level predictions, such as pooling or output MLP, are omitted for simplicity.

**Linear Transformers.** The *quadratic* complexity of attention in the original Transformer architecture [55] motivated the search for more efficient attention mechanisms that would scale *linearly* with the sequence length. Most of such *linear transformers* are developed for language modeling tasks, e.g., Linformer [58], Reformer [34], Longformer [4], Performer [12], BigBird [66], and have a dedicated Long Range Arena benchmark [52] to study the limits of models against extremely long input sequences. Pyraformer [40] is an example of a linear transformer for time series data, whereas S4 [23] is a more general signal processing approach that employs the state space model theory without the attention mechanism. In the graph domain, linear transformers are not well studied. Choromanski et al. [11] are the first to adapt Performer-style attention kernelization to small graphs.

## 3 Methods

In this work we provide a general, powerful, scalable (GPS) architecture for graph Transformers, following our 3-part recipe presented in Figure 1. We begin by categorization of existing positional (PE) and structural encodings (SE), a necessary ingredient for graph Transformers. Next, we analyse how these encodings also increase expressive power of MPNNs. The increased expressivity thus provides double benefit to our hybrid MPNN+Transformer architecture, which we introduce in Section 3.3. Last but not least, we provide an extensible implementation of GPS in GRAPHGPS package, built on top of PyG [20] and GraphGym [65].

### 3.1 Modular positional and structural encodings

One of our contribution is to provide a modular framework for PE/SE. It was shown in previous works that they are one of the most important factors in driving the performance of graph Transformers. Thus, a better understanding and organization of the PE and SE will aid in building of a more modular architecture and in guiding of the future research.

We propose to organize the PE and SE into 3 categories: **local**, **global** and **relative** in order to facilitate the integration within the pipeline and facilitate new research directions. They are presented visually in Figure 1, with more details in Table 1. Although PE and SE can appear similar to some

Table 1: The proposed categorization of positional encodings (PE) and structural encodings (SE). Some encodings are assigned to multiple categories in order to show their multiple expected roles.

| Encoding type | Description | Examples |
|---|---|---|
| **Local PE** *node features* | Allow a node to know its position and role within a local cluster of nodes. *Within a **cluster**, the closer two nodes are to each other, the closer their local PE will be, such as the position of a word in a sentence (not in the text).* | • Sum each column of non-diagonal elements of the $m$-steps random walk matrix. • Distance between a node and the centroid of a cluster containing the node. |
| **Global PE** *node features* | Allow a node to know its global position within the graph. *Within a **graph**, the closer two nodes are, the closer their global PE will be, such as the position of a word in a text.* | • Eigenvectors of the Adjacency, Laplacian [15, 36] or distance matrices. • SignNet [39] (includes aspects of relative PE and local SE). • Distance from the graph's centroid. • Unique identifier for each connected component of the graph. |
| **Relative PE** *edge features* | Allow two nodes to understand their distances or directional relationships. *Edge embedding that is correlated to the distance given by any global or local PE, such as the distance between two words.* | • Pair-wise node distances [38, 3, 36, 63, 44] based on shortest-paths, heat kernels, random-walks, Green's function, graph geodesic, or any local/global PE. • Gradient of eigenvectors [3, 36] or any local/global PE. • PEG layer [57] with specific node-wise distances. • Boolean indicating if two nodes are in the same cluster. |
| **Local SE** *node features* | Allow a node to understand what sub-structures it is a part of. *Given an SE of radius $m$, the more similar the $m$-hop subgraphs around two nodes are, the closer their local SE will be.* | • Degree of a node [63]. • Diagonal of the $m$-steps random-walk matrix [16]. • Time-derivative of the heat-kernel diagonal (gives the degree at $t = 0$). • Enumerate or count predefined structures such as triangles, rings, etc. [6, 68]. • Ricci curvature [54]. |
| **Global SE** *graph features* | Provide the network with information about the global structure of the graph. *The more similar two graphs are, the closer their global SE will be.* | • Eigenvalues of the Adjacency or Laplacian matrices [36]. • Graph properties: diameter, girth, number of connected components, # of nodes, # of edges, nodes-to-edges ratio. |
| **Relative SE** *edge features* | Allow two nodes to understand how much their structures differ. *Edge embedding that is correlated to the difference between any local SE.* | • Pair-wise distance, encoding, or gradient of any local SE. • Boolean indicating if two nodes are in the same sub-structure [5] (similar to the gradient of sub-structure enumeration). |

extent, they are different yet complementary. PE gives a notion of distance, while SE gives a notion of structural similarity. One can always infer certain notions of distance from large structures, or certain notions of structure from short distances, but this is not a trivial task, and the objective of providing PE and SE remains distinct, as discussed in the following subsections.

Despite presenting a variety of possible functions, we focus our empirical evaluations on the **global PE**, **relative PE** and **local SE** since they are known to yield significant improvements. We leave the empirical evaluation of other encodings for future work.

**Positional encodings (PE)** are meant to provide an idea of the *position in space* of a given node within the graph. Hence, when two nodes are close to each other within a graph or subgraph, their PE should also be close. A common approach is to compute the pair-wise distance between each pairs of nodes or their eigenvectors as proposed in [38, 63, 36, 57], but this is not compatible with linear Transformers as it requires to materialize the full attention matrix [12]. Instead, we want the PE to either be features of the nodes or real edges of the graph, thus a better fitting solution is to use the eigenvectors of the graph Laplacian or their gradient [15, 3, 36]. See Table 1 for more PE examples.

**Structural encodings (SE)** are meant to provide an embedding of the structure of graphs or subgraphs to help increase the expressivity and the generalizability of graph neural networks (GNN). Hence, when two nodes share similar subgraphs, or when two graphs are similar, their SE should also be close. Simple approaches are to identify pre-defined patterns in the graphs as one-hot encodings, but they require expert knowledge of graphs [6, 5]. Instead, using the diagonal of the $m$-steps random-walk matrix encodes richer information into each node [16], such as for odd $m$ it can indicate if a node is a part of an $m$-long cycle. Structural encodings can also be used to define the *global* graph structure, for instance using the eigenvalues of the Laplacian, or as *relative* edge features to identify if nodes are contained within the same clusters, with more examples in Table 1.

## 3.2 Why do we need PE and SE in MPNN?

As reviewed earlier, several recent GNNs make use of positional encodings (PE) and structural encodings (SE) as soft biases to improve the model expressivity (summarized in Table 1), which also leads to better generalization. In this section, we present an examination of PE and SE by showing how message-passing networks, despite operating on the graph structure, remain blind to the information encapsulated by the PE and SE.

**1-Weisfeiler-Leman test (1-WL).** It is well known that standard MPNNs are as expressive as the 1-WL test, meaning that they fail to distinguish non-isomorphic graphs under a 1-hop aggregation. We argue that the selected *local*, *global* and *relative* PE/SE allow MPNNs to become more expressive than the 1-WL test, thus making them fundamentally more expressive at distinguishing between nodes and graphs. To this end, we study the following two types of graphs (Figure 2 and Appendix C.1).

**Circular Skip Link (CSL) graph.** In a CSL graph-pair [46], we want to be able to distinguish the two non-isomorphic graphs. Since the 1-WL algorithm produces the same color for every node in both graphs, also every MPNN will fail to distinguish them. However, using a *global* PE (e.g., Laplacian PE [15]) assigns each node a unique initial color and makes the CSL graph-pair distinguishable. This demonstrates that an MPNN cannot learn such a PE from the graph structure alone. Next, using a *local* SE (e.g., diagonals of $m$-steps random walk) can successfully capture the difference in the skip links of the two graphs [42], resulting in their different node coloring [16].

**Decalin molecule.** In the bicyclic Decalin graph, Figures 2b and C.1b, the node $a$ is isomorphic to node $b$, and so is the node $c$ to node $d$. A 1-WL coloring of the nodes, and analogously MPNN, would generate one color for the nodes $a, b$ and another color for $c, d$. The same applies to the aforementioned *local* SE [16]. In case of link prediction, this causes potential links $(a, d)$ and $(b, d)$ to be indistinguishable [67]. Using a distance-based *relative* PE on the edges or an eigenvector-based *global* PE, however, would allow to differentiate the two links.

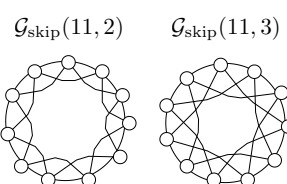

$\mathcal{G}_{\text{skip}}(11, 2)$   $\mathcal{G}_{\text{skip}}(11, 3)$

(a) Circular Skip Link graphs

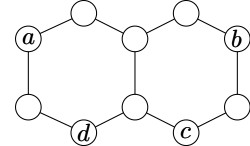

(b) Decalin molecular graph

Figure 2: Example graphs with anonymous nodes without distinguishing features.

## 3.3 GPS layer: an MPNN+Transformer hybrid

In this section we introduce the GPS layer, which is a hybrid MPNN+Transformer layer. First we argue how it alleviates the limitation of a closely related work. Next, we list the layer update equations which can be instantiated with a variety of MPNN and Transformer layers. Finally, we present its characteristics in terms of modularity, scalability and expressivity.

**Preventing early smoothing.** Why not use an architecture like GraphTrans [31] comprising of a few layers of MPNNs before the Transformer? Since MPNNs are limited by problems of over-smoothing, over-squashing, and low expressivity against the WL test [1, 54], these layers could *irreparably* fail to keep some information in the early stage. Although they could make use of PE/SE or more expressive MPNNs [3, 16], they are still likely to lose information. An analogous 2-stage strategy was successful in computer vision [18, 24] thanks to the high expressivity of convolutional layers on grids, but we do not expect it to achieve the same success on graphs due to the limitations of message-passing.

**Update function.** At each layer, the features are updated by aggregating the output of an MPNN layer with that of a global attention layer, as shown in Figures 1 and D.1, and described by the equations below. Note that the edge features are only passed to the `MPNN` layer, and that residual connections with batch normalization [30] are omitted for clarity. Both the `MPNN` and `GlobalAttn` layers are modular, i.e., `MPNN` can be any function that acts on a local neighborhood and `GlobalAttn` can be any fully-connected layer.

$$\mathbf{X}^{\ell+1}, \mathbf{E}^{\ell+1} = \texttt{GPS}^{\ell}\left(\mathbf{X}^{\ell}, \mathbf{E}^{\ell}, \mathbf{A}\right) \tag{1}$$

$$\text{computed as} \quad \mathbf{X}_M^{\ell+1}, \mathbf{E}^{\ell+1} = \texttt{MPNN}_e^{\ell}\left(\mathbf{X}^{\ell}, \mathbf{E}^{\ell}, \mathbf{A}\right), \tag{2}$$

$$\mathbf{X}_T^{\ell+1} = \texttt{GlobalAttn}^{\ell}\left(\mathbf{X}^{\ell}\right), \tag{3}$$

$$\mathbf{X}^{\ell+1} = \texttt{MLP}^{\ell}\left(\mathbf{X}_M^{\ell+1} + \mathbf{X}_T^{\ell+1}\right), \tag{4}$$

where $\mathbf{A} \in \mathbb{R}^{N \times N}$ is the adjacency matrix of a graph with $N$ nodes and $E$ edges; $\mathbf{X}^\ell \in \mathbb{R}^{N \times d_\ell}$, $\mathbf{E}^\ell \in \mathbb{R}^{E \times d_\ell}$ are the $d_\ell$-dimensional node and edge features, respectively; $\texttt{MPNN}_e^\ell$ and $\texttt{GlobalAttn}^\ell$ are instances of an MPNN with edge features and of a global attention mechanism at the $\ell$-th layer with their corresponding learnable parameters, respectively; $\texttt{MLP}^\ell$ is a 2-layer MLP block.

**Modularity** is achieved by allowing drop-in replacement for a number of module choices, including the initial PE/SE types, the networks that processes those PE/SE, the MPNN and global attention layers that constitute a GPS layer, and the final task-specific prediction head. Further, as research advances in different directions, GRAPHGPS allows to easily implement new PE/SE and other layers.

**Scalability** is achieved by allowing for a computational complexity linear in both the number of nodes and edges $O(N + E)$; excluding the potential precomputation step required for various PE, such as Laplacian eigen-decomposition. By restricting the PE/SE to real nodes and edges, and by excluding the edge features from the global attention layer, we can avoid materializing the full quadratic attention matrix. Therefore we can utilize a linear Transformer with $O(N)$ complexity, while the complexity of an MPNN is $O(E)$. For sparse graphs such as molecular graphs, regular graphs, and knowledge graphs, the edges are practically proportional to the nodes $E = \Theta(N)$, meaning the entire complexity can be considered linear in the number of nodes $O(N)$. Empirically, even on small molecular graphs, our architecture reduces computation time compared to other GT models, e.g., a model of ~6M parameters requires 196s per epoch on the ogbg-molpcba [27] dataset, compared to 883s for SAN [36] on the same GPU type.

**Expressivity** in terms of sub-structure identification and the Weisfeiler-Leman (WL) test is achieved via providing a rich set of PE/SE, as proposed in various works [3, 36, 16, 5, 6] and detailed in Section 3.1. Further, the Transformer allows to resolve the expressivity bottlenecks caused by over-smoothing [36] and over-squashing [1] by allowing information to spread across the graph via full-connectivity. Finally, in Section 3.4, we demonstrate that, given the right components, the proposed architecture does not lose edge information and is a universal function approximator on graphs.

## 3.4 Theoretical expressivity

In this section, we first discuss how the MPNN layer allows to propagate edge and neighbor information on the nodes. Then, we show that the proposed model is a universal function approximator on graphs, similarly to the SAN architecture [36].

**Preserving edge information in the Transformer layer.** Most GTs do not fully utilize edge features of the input graph. The Graph Transformer [14], SAN [36] and Graphormer [63] only use edge features to condition the attention between a pair of nodes, that is, they influence the attention gating mechanism but are not explicitly involved in updating of the node representations. GraphiT [44] does not consider edge features at all. Recent 2-step methods GraphTrans [31] and SAT [9] can use edge features in their first MPNN step, however this step is applied only once and typically includes several $k$ rounds of message passing. Therefore this latter approach may suffer from initial over-smoothing, as $k$-hop neighborhoods together with the respective edge features need to be represented in a fixed-sized node representation.

On the other hand, in GPS, interleaving one round of local neighborhood aggregation via an MPNN layer with global self-attention mechanism reduces the initial representation bottleneck and enables iterative local and global interactions. In the attention, the key-query-value mechanism only explicitly depends on the node features, but assuming efficient representation encoding by the MPNN, the node features can implicitly encode edge information, thus edges can play a role in either the key, query, or values. In Appendix C.2, we give a more formal argument on how, following an MPNN layer, node features can encode edge features alongside information related to node-connectivity.

**Universal function approximator on graphs.** Kreuzer et al. [36][Sec. 3.5] demonstrated the universality of graph Transformers. It was shown that, given the full set of Laplacian eigenvectors, the model was a universal function approximator on graphs and could provide an approximate solution to the isomorphism problem, making it more powerful than any Weisfeiler-Leman (WL) isomorphism test given enough parameters. Here, we argue that the same holds for our architecture since we can also use the full set of eigenvectors, and since all edge information can be propagated to the nodes.

Table 2: Summary of the ablation studies. Details of the architectural choices, parameters, standard deviation, and computation times are presented in Appendix B.

(a) Ablation of the Transformer and MPNN layers. We observe a major drop when using only a Transformer without an MPNN. Further, most datasets benefit from using a Transformer, without any negative impact.

| | Ablation | ZINC | PCQM4Mv2 subset | CIFAR10 | MalNet -Tiny |
|---|---|---|---|---|---|
| | | MAE ↓ | MAE ↓ | Acc. ↑ | Acc. ↑ |
| **Global Attention** | *none* | 0.070 | 0.1213 | 69.95 | 92.23 |
| | Transformer | 0.070 | 0.1159 | 72.31 | 93.50 |
| | Performer | 0.071 | 0.1142 | 70.67 | 92.64 |
| | BigBird | 0.071 | 0.1237 | 70.48 | 92.34 |
| **MPNN** | *none* | 0.217 | 0.3294 | 68.86 | 73.90 |
| | GINE | 0.070 | 0.1284 | 71.11 | 92.27 |
| | GatedGCN | 0.086 | 0.1159 | 72.31 | 92.64 |
| | PNA | 0.070 | 0.1409 | 73.42 | 91.67 |

(b) Ablation of the PE and SE types. RWSE provides consistent gains at relatively low computational cost, while SignNet$^{DeepSets}$ is the single best performing encoding, albeit at increased computational cost.

| | Ablation | ZINC | PCQM4Mv2 subset | CIFAR10 | MalNet -Tiny |
|---|---|---|---|---|---|
| | | MAE ↓ | MAE ↓ | Acc. ↑ | Acc. ↑ |
| **PE / SE** | *none* | 0.113 | 0.1355 | 71.49 | 92.64 |
| | RWSE | 0.070 | 0.1159 | 71.96 | 92.77 |
| | LapPE | 0.116 | 0.1201 | 72.31 | 92.74 |
| | SignNet$^{MLP}$ | 0.090 | 0.1158 | 71.74 | 92.57 |
| | SignNet$^{DeepSets}$ | 0.079 | 0.1144 | 72.37 | 93.13 |
| | PEG$^{LapEig}$ | 0.161 | 0.1209 | 72.10 | 92.27 |

*Encodings are color-coded by their positional or structural type.

## 4   Experiments

We perform ablation studies on 4 datasets to evaluate the contribution of the message-passing module, the global attention module, and the positional or structural encodings. Then, we evaluate GPS on a diverse set of 11 benchmarking datasets, and show state-of-the-art (SOTA) results in many cases.

We test on datasets from different sources to ensure diversity, providing their detailed description in Appendix A.1. From the Benchmarking GNNs [15], we test on the ZINC, PATTERN, CLUSTER, MNIST, CIFAR10. From the open graph benchmark (OGB) [27], we test on all graph-level datasets: ogbg-molhiv, ogbg-molpcba, ogbg-code2, and ogbg-ppa, and from their large-scale challenge we test on the OGB-LSC PCQM4Mv2 [28]. Finally, we also select MalNet-Tiny [21] with 5000 graphs, each of up to 5000 nodes, since the number of nodes provide a scaling challenge for Transformers.

### 4.1   Ablation studies

In this section, we evaluate multiple options for the three main components of our architecture in order to gauge their contribution to predictive performance and to better guide dataset-specific hyper-parameter optimization. First, we quantify benefits of the considered global-attention modules in 4 tasks. Then, we note that the MPNN layer is an essential part for high-performing models, and identify the layer type most likely to help. Finally, we observe when different global PE or local SE provide significant boost in the performance. All ablation results are averaged over multiple random seeds and summarized in Table 2, with additional information available in Appendix B.

**Global-Attention module.**   Here we consider global attention implemented as $O(N^2)$ key-query-value Transformer attention or linear-time attention mechanisms of Performer or BigBird. We notice in Table 2a that using a Transformer is always beneficial, except for the ZINC dataset where no changes are observed. This motivates our architecture and the hypothesis that long-range dependencies are generally important. We further observe that Performer falls behind Transformer in terms of the predictive performance, although it provides a gain over the baseline and the ability to scale to very large graphs. Finally, BigBird in our setting offers no significant gain, while also being slower than Performer (see Appendix B).

Having no gain on the ZINC dataset is expected since the task is a combination of the computed octanol-water partition coefficient (cLogP) [60] and the synthetic accessibility score (SA-score) [19], both of which only count occurrences of local sub-structures. Hence, there is no need for a global connectivity, but a strong need for structural encodings.

**Message-passing module.**   Next, we evaluate the effect of various message-passing architectures, Table 2a. It is apparent that they are fundamental to the success of our method: removing the layer leads to a significant drop in performance across all datasets. Indeed, without an MPNN, the edge

Table 3: Test performance in five benchmarks from [15]. Shown is the mean ± s.d. of 10 runs with different random seeds. Highlighted are the top **first**, **second**, and **third** results.

| Model | ZINC | MNIST | CIFAR10 | PATTERN | CLUSTER |
|---|---|---|---|---|---|
| | MAE ↓ | Accuracy ↑ | Accuracy ↑ | Accuracy ↑ | Accuracy ↑ |
| GCN [33] | 0.367 ± 0.011 | 90.705 ± 0.218 | 55.710 ± 0.381 | 71.892 ± 0.334 | 68.498 ± 0.976 |
| GIN [61] | 0.526 ± 0.051 | 96.485 ± 0.252 | 55.255 ± 1.527 | 85.387 ± 0.136 | 64.716 ± 1.553 |
| GAT [56] | 0.384 ± 0.007 | 95.535 ± 0.205 | 64.223 ± 0.455 | 78.271 ± 0.186 | 70.587 ± 0.447 |
| GatedGCN [7, 15] | 0.282 ± 0.015 | 97.340 ± 0.143 | 67.312 ± 0.311 | 85.568 ± 0.088 | 73.840 ± 0.326 |
| GatedGCN-LSPE [16] | 0.090 ± 0.001 | – | – | – | – |
| PNA [13] | 0.188 ± 0.004 | 97.94 ± 0.12 | 70.35 ± 0.63 | – | – |
| DGN [3] | 0.168 ± 0.003 | – | 72.838 ± 0.417 | 86.680 ± 0.034 | – |
| GSN [6] | 0.101 ± 0.010 | – | – | – | – |
| CIN [5] | 0.079 ± 0.006 | – | – | – | – |
| CRaWl [53] | 0.085 ± 0.004 | 97.944 ± 0.050 | 69.013 ± 0.259 | – | – |
| GIN-AK+ [68] | 0.080 ± 0.001 | – | 72.19 ± 0.13 | 86.850 ± 0.057 | – |
| SAN [36] | 0.139 ± 0.006 | – | – | 86.581 ± 0.037 | 76.691 ± 0.65 |
| Graphormer [63] | 0.122 ± 0.006 | – | – | – | – |
| K-Subgraph SAT [9] | 0.094 ± 0.008 | – | – | 86.848 ± 0.037 | 77.856 ± 0.104 |
| EGT [29] | 0.108 ± 0.009 | 98.173 ± 0.087 | 68.702 ± 0.409 | 86.821 ± 0.020 | 79.232 ± 0.348 |
| GPS (ours) | 0.070 ± 0.004 | 98.051 ± 0.126 | 72.298 ± 0.356 | 86.685 ± 0.059 | 78.016 ± 0.180 |

features are not taken into consideration at all. Additionally, without reinforcing of the local graph structure, the network can overfit to the PE/SE. This reiterates findings of Kreuzer et al. [36], where considerably larger weights were assigned to the local attention.

We also find that although a vanilla PNA [13] generally outperforms GINE [26] and GatedGCN [7], adding the PE and SE results in major performance boost especially for the GatedGCN. This is consistent with results of Dwivedi et al. [16] and shows the importance of these encodings for gating.

Perhaps the necessity of a local message-passing module is due to the limited amount of graph data, and scaling to colossal datasets [49] that we encounter in language and vision could change that. Indeed, the Graphormer architecture [63] was able to perform very well on the full PCQM4Mv2 dataset without a local module. However, even large Transformer-based language models [8] and vision models [25] can benefit from an added local aggregation and outperform pure Transformers.

**Positional/Structural Encodings.** Finally, we evaluate the effects of various PE/SE schemes, Table 2b. We find them generally beneficial to downstream tasks, in concordance to the vast literature on the subject (see Table 1). The benefits of the different encodings are very dataset dependant, with the random-walk structural encoding (RWSE) being more beneficial for molecular data and the Laplacian eigenvectors encodings (LapPE) being more beneficial in image superpixels. However, using SignNet with DeepSets encoding [39] as an improved way of processing the LapPE seems to be consistently successful across tasks. We hypothesize that SignNet can learn structural representation using the eigenvectors, for example, to generate local heat-kernels that approximate random walks [2]. Last but not least we evaluate PEG-layer design [57] with Laplacian eigenmap.

## 4.2 Benchmarking GPS

We compare GPS against a set of popular message-passing neural networks (GCN, GIN, GatedGCN, PNA, etc.), graph transformers (SAN, Graphormer, etc.), and other recent graph neural networks with SOTA results (CIN, CRaWL, GIN-AK+, ExpC). To ensure diverse benchmarking tasks, we use datasets from Benchmarking-GNNs [15], OGB [27] and its large-scale challenge [28], and Long-Range Graph Benchmark [17], with more details given in Appendix A.1. We report the mean and standard deviation over 10 random seeds if not explicitly stated otherwise.

**Benchmarking GNNs [15].** We first benchmark our method on 5 tasks from Benchmarking GNNs [15], namely ZINC, MNIST, CIFAR10, PATTERN, and CLUSTER, shown in Table 3. We observe that our GPS gives SOTA results on ZINC and the second best in 3 more datasets, showcasing the ability to perform very well on a variety of synthetic tasks designed to test the model expressivity.

**Open Graph Benchmark [27].** Next, we benchmark on all 4 graph-level tasks from OGB, namely molhiv, molpcba, ppa, and code2, Table 4. On the molhiv dataset, we observed our model to suffer

Table 4: Test performance in graph-level OGB benchmarks [27]. Shown is the mean ± s.d. of 10 runs. Models that were first pre-trained on another dataset or use an ensemble are not included here.

| Model | ogbg-molhiv | ogbg-molpcba | ogbg-ppa | ogbg-code2 |
|---|---|---|---|---|
| | AUROC ↑ | Avg. Precision ↑ | Accuracy ↑ | F1 score ↑ |
| GCN+virtual node | 0.7599 ± 0.0119 | 0.2424 ± 0.0034 | 0.6857 ± 0.0061 | 0.1595 ± 0.0018 |
| GIN+virtual node | 0.7707 ± 0.0149 | 0.2703 ± 0.0023 | 0.7037 ± 0.0107 | 0.1581 ± 0.0026 |
| GatedGCN-LSPE | – | 0.267 ± 0.002 | – | – |
| PNA | 0.7905 ± 0.0132 | 0.2838 ± 0.0035 | – | 0.1570 ± 0.0032 |
| DeeperGCN | 0.7858 ± 0.0117 | 0.2781 ± 0.0038 | 0.7712 ± 0.0071 | – |
| DGN | 0.7970 ± 0.0097 | 0.2885 ± 0.0030 | – | – |
| GSN (directional) | 0.8039 ± 0.0090 | – | – | – |
| GSN (GIN+VN base) | 0.7799 ± 0.0100 | – | – | – |
| CIN | 0.8094 ± 0.0057 | – | – | – |
| GIN-AK+ | 0.7961 ± 0.0119 | 0.2930 ± 0.0044 | – | – |
| CRaWl | – | 0.2986 ± 0.0025 | – | – |
| ExpC [62] | 0.7799 ± 0.0082 | 0.2342 ± 0.0029 | 0.7976 ± 0.0072 | – |
| SAN | 0.7785 ± 0.2470 | 0.2765 ± 0.0042 | – | – |
| GraphTrans (GCN-Virtual) | – | 0.2761 ± 0.0029 | – | 0.1830 ± 0.0024 |
| K-Subtree SAT | – | – | 0.7522 ± 0.0056 | 0.1937 ± 0.0028 |
| GPS (ours) | 0.7880 ± 0.0101 | 0.2907 ± 0.0028 | 0.8015 ± 0.0033 | 0.1894 ± 0.0024 |

Table 5: Evaluation on PCQM4Mv2 [28] dataset. For GPS evaluation, we treated the *validation* set of the dataset as a test set, since the *test-dev* set labels are private. For more details refer to Appendix A.

| Model | PCQM4Mv2 | | | |
|---|---|---|---|---|
| | Test-dev MAE ↓ | Validation MAE ↓ | Training MAE | # Param. |
| GCN | 0.1398 | 0.1379 | n/a | 2.0M |
| GCN-virtual | 0.1152 | 0.1153 | n/a | 4.9M |
| GIN | 0.1218 | 0.1195 | n/a | 3.8M |
| GIN-virtual | 0.1084 | 0.1083 | n/a | 6.7M |
| GRPE [48] | 0.0898 | 0.0890 | n/a | 46.2M |
| EGT [29] | 0.0872 | 0.0869 | n/a | 89.3M |
| Graphormer [51] | n/a | 0.0864 | 0.0348 | 48.3M |
| GPS-small | n/a | 0.0938 | 0.0653 | 6.2M |
| GPS-medium | n/a | 0.0858 | 0.0726 | 19.4M |

from overfitting, but to still outperform SAN, while other graph Transformers do not report results. For the molpcba, ppa, and code2, GPS always ranks among the top 3 models, highlighting again the versatility and expressiveness of the GPS approach. Further, GPS outperforms every other GT on all 4 benchmarks, except SAT on code2.

**OGB-LSC PCQM4Mv2 [28].** The large-scale PCQM4Mv2 dataset has been a popular benchmark for recent GTs, particularly due to Graphormer [63] winning the initial challenge. We report the results in Table 5, observing significant improvements over message-passing networks at comparable parameter budget. GPS also outperforms GRPE [48], EGT [29], and Graphormer [63] with less than half their parameters, and with significantly less overfitting on the training set. Contrarily to Graphormer, we do not need to precompute spatial distances from approximate 3D molecular conformers [64], the RWSEs we utilize are graph-based only.

**MalNet-Tiny.** The MalNet-Tiny [21] dataset consists of function call graphs with up to 5,000 nodes. These graphs are considerably larger than previously considered inductive graph-learning benchmarks, which enables us to showcase scalability of GPS to much larger graphs than prior methods. Our GPS reaches $92.72\% \pm 0.7$pp test accuracy when using Performer global attention. Interestingly, using Transformer global attention leads to further improved GPS performance, $93.36\% \pm 0.6$pp (based on 10 runs), albeit at the cost of doubled run-time. In both cases, we used comparable architecture to Freitas et al. [21], with 5 layers and 64 dimensional hidden node representation, and outperform their best GIN model with $90\%$ accuracy. See Appendix B for GPS ablation study on MalNet-Tiny.

**Long-Range Graph Benchmark [17].** Finally, we evaluate the GPS method on a recent Long-Range Graph Benchmark (LRGB) suite of 5 datasets that are intended to test a method's ability to capture long-range dependencies in the input graphs. We abide to the ~500k model parameter budget and

Table 6: Test performance on long-range graph benchmarks (LRGB) [17]. Shown is the mean ± s.d. of 4 runs. The **first**, second, and third best are highlighted.
*SAN on COCO-SP exceeded 60h time limit on an NVidia A100 GPU system.

| Model | PascalVOC-SP | COCO-SP | Peptides-func | Peptides-struct | PCQM-Contact |
|---|---|---|---|---|---|
| | F1 score ↑ | F1 score ↑ | AP ↑ | MAE ↓ | MRR ↑ |
| GCN | 0.1268 ± 0.0060 | 0.0841 ± 0.0010 | 0.5930 ± 0.0023 | 0.3496 ± 0.0013 | 0.3234 ± 0.0006 |
| GINE | 0.1265 ± 0.0076 | 0.1339 ± 0.0044 | 0.5498 ± 0.0079 | 0.3547 ± 0.0045 | 0.3180 ± 0.0027 |
| GatedGCN | 0.2873 ± 0.0219 | 0.2641 ± 0.0045 | 0.5864 ± 0.0077 | 0.3420 ± 0.0013 | 0.3218 ± 0.0011 |
| GatedGCN+RWSE | 0.2860 ± 0.0085 | 0.2574 ± 0.0034 | 0.6069 ± 0.0035 | 0.3357 ± 0.0006 | 0.3242 ± 0.0008 |
| Transformer+LapPE | 0.2694 ± 0.0098 | 0.2618 ± 0.0031 | 0.6326 ± 0.0126 | 0.2529 ± 0.0016 | 0.3174 ± 0.0020 |
| SAN+LapPE | 0.3230 ± 0.0039 | 0.2592 ± 0.0158* | 0.6384 ± 0.0121 | 0.2683 ± 0.0043 | 0.3350 ± 0.0003 |
| SAN+RWSE | 0.3216 ± 0.0027 | 0.2434 ± 0.0156* | 0.6439 ± 0.0075 | 0.2545 ± 0.0012 | 0.3341 ± 0.0006 |
| GPS (ours) | 0.3748 ± 0.0109 | 0.3412 ± 0.0044 | 0.6535 ± 0.0041 | 0.2500 ± 0.0005 | 0.3337 ± 0.0006 |

closely follow the experimental setup and hyperparameter choices of the graph Transformer baselines tested in LRGB [17]. We keep the same node/edge encoders and model depth (number of layers), deviating only in two aspects: i) we slightly decrease the size of hidden node representations to fit within the parameter budget, ii) we employ cosine learning rate schedule as in our other experiments (Section A.3). For each dataset we utilize LapPE positional encodings and GPS with GatedGCN [7] and Transformer [55] components.

GPS improves over all evaluated baselines in 4 out of 5 LRGB datasets (Table 6). Additionally, we conducted GPS ablation studies on PascalVOC-SP and Peptides-func datasets in the same fashion as for 4 previous datasets in Table 2, presented in Tables B.5 and B.6, respectively. For both datasets the global attention, in form of Transformer or Performer, is shown to be a critical component of the GPS in outperforming MPNNs. In the case of PascalVOC-SP, interestingly, the Laplacian PEs are not beneficial, as without them the GPS scores even higher F1-score $0.3846$, and PEG [57] relative distance embeddings enable the highest score of $0.3956$.

## 5   Conclusion

Our work is setting the foundation for a unified architecture of graph neural networks, with modular and scalable graph Transformers and a broader understanding of the role of graphs with positional and structural encodings. In our ablation studies, we demonstrated the importance of each module: the Transformer, flexible message-passing, and rich positional and structural encodings all contributed to the success of GPS on a wide variety of benchmarks. Indeed, considering 5 Benchmarking-GNN tasks [15], 5 OGB(-LSC) tasks [27, 28], 5 LRGB tasks [17] and MalNet-Tiny, we outperformed every graph Transformer on 11 out of 16 tasks while also achieving state-of-the-art on 8 of them. We further showed that the model can scale to very large graphs of several thousand nodes, far beyond any previous graph Transformer. By open-sourcing the GRAPHGPS package, we hope to accelerate the research in efficient and expressive graph Transformers, and move the field closer to a unified hybrid Transformer architecture for graphs.

**Limitations.** We find that graph transformers are sensitive to hyperparameters and there is no *one-size-fits-all* solution for all datasets. We also identify a lack of challenging graph datasets necessitating long-range dependencies where linear attention architectures could exhibit all scalability benefits.

**Societal Impact.** As a general graph representation learning method, we do not foresee immediate negative societal outcomes. However, its particular application, e.g., in drug discovery or computational biology, will have to be thoroughly examined for trustworthiness or malicious usage.

## Acknowledgments and Disclosure of Funding

This work was partially funded by IVADO (Institut de valorisation des données) grant PRF-2019-3583139727 and Canada CIFAR AI Chair [*G.W.*]. This research is supported by Nanyang Technological University, under SUG Grant (020724-00001) [*V.P.D.*] and Samsung AI graph at Mila [*M.G.*]. The content provided here is solely the responsibility of the authors and does not necessarily represent the official views of the funding agencies.

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
