# A  Experimental Details

## A.1  Datasets description

Table A.1: Overview of the graph learning dataset [15, 27, 28, 21, 17] used in this study.

| Dataset | # Graphs | Avg. # nodes | Avg. # edges | Directed | Prediction level | Prediction task | Metric |
|---|---|---|---|---|---|---|---|
| ZINC | 12,000 | 23.2 | 24.9 | No | graph | regression | Mean Abs. Error |
| MNIST | 70,000 | 70.6 | 564.5 | Yes | graph | 10-class classif. | Accuracy |
| CIFAR10 | 60,000 | 117.6 | 941.1 | Yes | graph | 10-class classif. | Accuracy |
| PATTERN | 14,000 | 118.9 | 3,039.3 | No | inductive node | binary classif. | Accuracy |
| CLUSTER | 12,000 | 117.2 | 2,150.9 | No | inductive node | 6-class classif. | Accuracy |
| ogbg-molhiv | 41,127 | 25.5 | 27.5 | No | graph | binary classif. | AUROC |
| ogbg-molpcba | 437,929 | 26.0 | 28.1 | No | graph | 128-task classif. | Avg. Precision |
| ogbg-ppa | 158,100 | 243.4 | 2,266.1 | No | graph | 37-task classif. | Accuracy |
| ogbg-code2 | 452,741 | 125.2 | 124.2 | Yes | graph | 5 token sequence | F1 score |
| PCQM4Mv2 | 3,746,620 | 14.1 | 14.6 | No | graph | regression | Mean Abs. Error |
| MalNet-Tiny | 5,000 | 1,410.3 | 2,859.9 | Yes | graph | 5-class classif. | Accuracy |
| PascalVOC-SP | 11,355 | 479.4 | 2,710.5 | No | inductive node | 21-class classif. | F1 score |
| COCO-SP | 123,286 | 476.9 | 2,693.7 | No | inductive node | 81-class classif. | F1 score |
| PCQM-Contact | 529,434 | 30.1 | 61.0 | No | inductive link | link ranking | MRR |
| Peptides-func | 15,535 | 150.9 | 307.3 | No | graph | 10-task classif. | Avg. Precision |
| Peptides-struct | 15,535 | 150.9 | 307.3 | No | graph | 11-task regression | Mean Abs. Error |

**ZINC** [15] (MIT License) consists of 12K molecular graphs from the ZINC database of commercially available chemical compounds. These molecular graphs are between 9 and 37 nodes large. Each node represents a heavy atom (28 possible atom types) and each edge represents a bond (3 possible types). The task is to regress constrained solubility (logP) of the molecule. The dataset comes with a predefined 10K/1K/1K train/validation/test split.

**MNIST and CIFAR10** [15] (CC BY-SA 3.0 and MIT License) are derived from like-named image classification datasets by constructing an 8 nearest-neighbor graph of SLIC superpixels for each image. The 10-class classification tasks and standard dataset splits follow the original image classification datasets, i.e., for MNIST 55K/5K/10K and for CIFAR10 45K/5K/10K train/validation/test graphs.

**PATTERN and CLUSTER** [15] (MIT License) are synthetic datasets sampled from Stochastic Block Model. Unlike other datasets, the prediction task here is an inductive node-level classification. In PATTERN the task is to recognize which nodes in a graph belong to one of 100 possible sub-graph patterns that were randomly generated with different SBM parameters than the rest of the graph. In CLUSTER, every graph is composed of 6 SBM-generated clusters, each drawn from the same distribution, with only a single node per cluster containing a unique cluster ID. The task is to infer which cluster ID each node belongs to.

**ogbg-molhiv and ogbg-molpcba** [27] (MIT License) are molecular property prediction datasets adopted by OGB from MoleculeNet. These datasets use a common node (atom) and edge (bond) featurization that represent chemophysical properties. The prediction task of ogbg-molhiv is binary classification of molecule's fitness to inhibit HIV replication. The ogbg-molpcba, derived from PubChem BioAssay, targets to predict results of 128 bioassays in multi-task binary classification setting.

**ogbg-ppa** [27] (CC-0 license) consists of protein-protein association (PPA) networks derived from 1581 species categorized to 37 taxonomic groups. Nodes represent proteins and edges encode the normalized level of 7 different associations between two proteins. The task is to classify which of the 37 groups does a PPA network originate from.

**ogbg-code2** [27] (MIT License) is comprised of abstract syntax trees (ASTs) derived from source code of functions written in Python. The task is to predict the first 5 subtokens of the original function's name.

A small number of these ASTs are much larger than the average size in the dataset. Therefore we truncated ASTs with over 1000 nodes and kept the first 1000 nodes according to their depth in the AST. This impacted 2521 (0.5%) graphs in the dataset.

**OGB-LSC PCQM4Mv2** [28] (CC BY 4.0 license) is a large-scale molecular dataset that shares the same featurization as ogbg-mol* datasets. The task is to regress the HOMO-LUMO gap, a quantum physical property originally calculated using Density Functional Theory. True labels for original "test-dev" and "test-challange" dataset splits are kept private by the OGB-LSC challenge organizers. Therefore for the purpose of this paper we used the original *validation* set as the test set, while we left out random 150K molecules for our validation set.

**PCQM4Mv2-Subset** (under the original PCQM4Mv2 CC BY 4.0 license) is a subset of PCQM4Mv2 [28] that we created for the purpose of our ablation study. We sub-sampled the above-mentioned version of PCQM4Mv2 as follows; training set: 10%; validation set: 33%; test set: unchanged. This resulted in retaining 446,405 molecular graphs in total.

**MalNet-Tiny** [21] (CC-BY license) is a subset of MalNet that is comprised of function call graphs (FCGs) derived from Android APKs. This subset contains 5,000 graphs of up to 5,000 nodes, each coming from benign software or 4 types of malware. The FCGs are stripped of any original node or edge features, the task is to predict the type of the software based on the structure alone. The benchmarking version of this dataset typically uses Local Degree Profile as the set of node features.

**PascalVOC-SP and COCO-SP** [17] (Custom license for Pascal VOC 2011 respecting Flickr terms of use, and CC BY 4.0 license) are derived by SLIC superpixelization of Pascal VOC and MS COCO image datasets. Both are node classification datasets, where each superpixel node belongs to a particular object class.

**PCQM-Contact** [17] (CC BY 4.0) is derived from PCQM4Mv2 and respective 3D molecular structures. The task is a binary link prediction, identifying pairs of nodes that are considered to be in 3D contact (<3.5Å) yet distant in the 2D graph (>5 hops). The default evaluation ranking metric used is the Mean Reciprocal Rank (MRR).

**Peptides-func and Peptides-struct** [17] (CC BY-NC 4.0) are both composed of atomic graphs of peptides retrieved from SATPdb. In Peptides-func the prediction is multi-label graph classification into 10 nonexclusive peptide functional classes. While for Peptides-struct the task is graph regression of 11 3D structural properties of the peptides.

## A.2   Dataset splits and random seeds

All evaluated benchmarks define a standard train/validation/test dataset split. We follow these and report mean performance and standard deviation from multiple execution runs with different random seeds.

All main benchmarking results are based on 10 executed runs, except PCQM4Mv2 (for which we show the result of a single random seed run) and LRGB (for which we use 4 seed). The OGB-LSC [28] leaderboard for PCQM4Mv2 does not keep track of variance w.r.t. random seeds. This is likely due to the size of the dataset, in our evaluation we had run 3 random seeds and the standard deviation for GPS-small was 0.00034 which is below the presentation precision.

For ablation studies we used a reduce number of 4 random seeds due to computational constraints, while for PCQM4Mv2-Subset and MalNet-Tiny we used 3 random seeds. All experiments in the ablation studies were run from scratch, results from the main text (with 10 repeats) were not reused.

## A.3   Hyperparameters

In our hyperparameter search, we experimented with a variety of positional and structural encodings, MPNN types, global attention mechanisms and their hyperparameters. Considering the large number of hyperparameters and datasets, we did not perform an exhaustive search or a grid search beyond the ablation studies presented in the main text, Section 4.1. We have extrapolated from those results and established the PE/SE type and layer types for the remaining datasets. For each dataset we then adjusted the number of layers, dimensionality $d^\ell$, and other remaining hyperparameters based on hyperparameters reported in the related literature, or eventually based on validation performance using

"line search" along one of the hyperparameters at a time. Namely, we followed several hyperparameter choices of SAN [36], SAT [9], Graphormer [63], and Freitas et al. [21].

For benchmarking datasets from Dwivedi et al. [15] we followed the most commonly used parameter budgets: up to 500k parameters for ZINC, PATTERN, and CLUSTER; and ~100k parameters for MNIST and CIFAR10.

The final hyperparameters are presented in Tables A.2, A.3, A.4, A.5, together with the number of parameters and median wall-clock run-time for node encoding precomputation, one full epoch (including validation and test split evaluation), and the total time spent in the main loop. See Section A.4 for more details on the run-time measurements.

In all our experiments we used AdamW [41] optimizer, with the default settings of $\beta_1 = 0.9$, $\beta_2 = 0.999$, and $\epsilon = 10^{-8}$, together with linear "warm-up" increase of the learning rate at the beginning of the training followed by its cosine decay. The length of the warm-up period, base learning rate, and the total number of epoch were adjusted per dataset and are listed together with other hyperparameters (Tables A.2, A.3, A.4, A.5).

Table A.2: GPS hyperparameters for five datasets from Dwivedi et al. [15].

| Hyperparameter | ZINC | MNIST | CIFAR10 | PATTERN | CLUSTER |
|---|---|---|---|---|---|
| # GPS Layers | 10 | 3 | 3 | 6 | 16 |
| Hidden dim | 64 | 52 | 52 | 64 | 48 |
| GPS-MPNN | GINE | GatedGCN | GatedGCN | GatedGCN | GatedGCN |
| GPS-GlobAttn | Transformer | Transformer | Transformer | Transformer | Transformer |
| # Heads | 4 | 4 | 4 | 4 | 8 |
| Dropout | 0 | 0 | 0 | 0 | 0.1 |
| Attention dropout | 0.5 | 0.5 | 0.5 | 0.5 | 0.5 |
| Graph pooling | sum | mean | mean | – | – |
| Positional Encoding | RWSE-20 | LapPE-8 | LapPE-8 | LapPE-16 | LapPE-10 |
| PE dim | 28 | 8 | 8 | 16 | 16 |
| PE encoder | linear | DeepSet | DeepSet | DeepSet | DeepSet |
| Batch size | 32 | 16 | 16 | 32 | 16 |
| Learning Rate | 0.001 | 0.001 | 0.001 | 0.0005 | 0.0005 |
| # Epochs | 2000 | 100 | 100 | 100 | 100 |
| # Warmup epochs | 50 | 5 | 5 | 5 | 5 |
| Weight decay | 1e-5 | 1e-5 | 1e-5 | 1e-5 | 1e-5 |
| # Parameters | 423,717 | 115,394 | 112,726 | 337,201 | 502,054 |
| PE precompute | 23s | 96s | 2.55min | 28s | 67s |
| Time (epoch/total) | 21s / 11.67h | 76s / 2.13h | 64s / 1.78h | 32s / 0.89h | 86s / 2.40h |

Table A.3: GPS hyperparameters for graph-level prediction datasets from OGB [27].

| Hyperparameter | ogbg-molhiv | ogbg-molpcba | ogbg-ppa | ogbg-code2 |
|---|---|---|---|---|
| # GPS Layers | 10 | 5 | 3 | 4 |
| Hidden dim | 64 | 384 | 256 | 256 |
| GPS-MPNN | GatedGCN | GatedGCN | GatedGCN | GatedGCN |
| GPS-GlobAttn | Transformer | Transformer | Performer | Performer |
| # Heads | 4 | 4 | 8 | 4 |
| Dropout | 0.05 | 0.2 | 0.1 | 0.2 |
| Attention dropout | 0.5 | 0.5 | 0.5 | 0.5 |
| Graph pooling | mean | mean | mean | mean |
| Positional Encoding | RWSE-16 | RWSE-16 | None | None |
| PE dim | 16 | 20 | – | – |
| PE encoder | linear | linear | – | – |
| Batch size | 32 | 512 | 32 | 32 |
| Learning Rate | 0.0001 | 0.0005 | 0.0003 | 0.0001 |
| # Epochs | 100 | 100 | 200 | 30 |
| # Warmup epochs | 5 | 5 | 10 | 2 |
| Weight decay | 1e-5 | 1e-5 | 1e-5 | 1e-5 |
| # Parameters | 558,625 | 9,744,496 | 3,434,533 | 12,454,066 |
| PE precompute | 58s | 8.33min | – | – |
| Time (epoch/total) | 96s / 2.64h | 196s / 5.44h | 276s / 15.33h | 1919s / 16h |

Table A.4: GPS hyperparameters for large-scale graph-level prediction dataset OGB-LSC PCQM4Mv2 [28] and MalNet-Tiny [21]. GPS-medium architecture follows several hyperparameter choices of Graphormer [63]. Listed run-times were measured on a single NVidia A100 GPU system.

| Hyperparameter | PCQM4Mv2 (GPS-small) | PCQM4Mv2 (GPS-medium) | MalNet-Tiny |
|---|---|---|---|
| # GPS Layers | 5 | 10 | 5 |
| Hidden dim | 304 | 384 | 64 |
| GPS-MPNN | GatedGCN | GatedGCN | GatedGCN |
| GPS-SelfAttn | Transformer | Transformer | Performer |
| # Heads | 4 | 16 | 4 |
| Dropout | 0 | 0.1 | 0 |
| Attention dropout | 0.5 | 0.1 | 0.5 |
| Graph pooling | mean | mean | max |
| Positional Encoding | RWSE-16 | RWSE-16 | None |
| PE dim | 20 | 20 | – |
| PE encoder | linear | linear | – |
| Batch size | 256 | 256 | 16 |
| Learning Rate | 0.0005 | 0.0002 | 0.0005 |
| # Epochs | 100 | 150 | 150 |
| # Warmup epochs | 5 | 10 | 10 |
| Weight decay | 0 | 0 | 1.00e-5 |
| # Parameters | 6,152,001 | 19,414,641 | 527,237 |
| PE precompute | 47min | 51min | – |
| Time (epoch/total) | 619s / 17.18h | 1124s / 46.82h | 46s / 1.92h |

Table A.5: GPS hyperparameters for 5 datasets from Long Range Graph Benchmark (LRGB) [17].

| Hyperparameter | PascalVOC-SP | COCO-SP | PCQM-Contact | Peptides-func | Peptides-struct |
|---|---|---|---|---|---|
| # GPS Layers | 4 | 4 | 4 | 4 | 4 |
| Hidden dim | 96 | 96 | 96 | 96 | 96 |
| GPS-MPNN | GatedGCN | GatedGCN | GatedGCN | GatedGCN | GatedGCN |
| GPS-SelfAttn | Transformer | Transformer | Transformer | Transformer | Transformer |
| # Heads | 8 | 8 | 4 | 4 | 4 |
| Dropout | 0 | 0 | 0 | 0 | 0 |
| Attention dropout | 0.5 | 0.5 | 0.5 | 0.5 | 0.5 |
| Graph pooling | – | – | – | mean | mean |
| Positional Encoding | LapPE-10 | LapPE-10 | LapPE-10 | LapPE-10 | LapPE-10 |
| PE dim | 16 | 16 | 16 | 16 | 16 |
| PE encoder | DeepSet | DeepSet | DeepSet | DeepSet | DeepSet |
| Batch size | 32 | 32 | 256 | 128 | 128 |
| Learning Rate | 0.0005 | 0.0005 | 0.0003 | 0.0003 | 0.0003 |
| # Epochs | 300 | 300 | 200 | 200 | 200 |
| # Warmup epochs | 10 | 10 | 10 | 5 | 5 |
| Weight decay | 0 | 0 | 0 | 0 | 0 |
| # Parameters | 510,453 | 516,273 | 512,704 | 504,362 | 504,459 |
| PE precompute | 8.7min | 1h 34min | 5.23min | 73s | 73s |
| Time (epoch/total) | 17.5s / 1.46h | 213s / 17.8h | 154s / 8.54h | 6.36s / 0.35h | 6.15s / 0.34h |

### A.4 Computing environment and used resources

Our implementation is based on PyG and its GraphGym module [20, 65] that are provided under MIT License. All experiments were run in a shared computing cluster environment with varying CPU and GPU architectures. These involved a mix of NVidia V100 (32GB), RTX8000 (48GB), and A100 (40GB) GPUs. The resource budget for each experiment was 1 GPU, between 4 and 6 CPUs, and up to 32GB system RAM. The only exception are ogbg-ppa and PCQM4Mv2 that due to their size required up to 48GB system RAM.

To measure the run-time we used Python `time.perf_counter()` function. Due to the variation in computing infrastructure and load on shared resources the execution time occasionally notably varied. Therefore for our ablation studies we used only compute nodes with NVidia A100 GPUs, which considerably improved the run-time consistency. We list the wall-clock run-time that is approximately a median of the observed durations.

# B  Detailed ablation studies

Here we present the detailed ablation studies on impact of various MPNN, self attention, and positional / structural encoding types on GPS performance and run-time. In each case, we varied a single part of the model at a time, keeping the rest of the GPS hyperparameters unchanged from the best selected architecture for a given dataset. Results on ZINC are shown in Table B.1, on PCQM4Mv2-Subset in Table B.2, on MalNet-Tiny in Table B.3, on CIFAR10 in Table B.4, on PascalVOC-SP in Table B.5, and on Peptides-func in Table B.6. The first data row of each table reproduces results of the best selected architecture with hyperparameters detailed in Appendix A; any deviations compared to the main benchmarking results of Section 4.2 are well within the reported standard deviation. While for benchmarking results we used 10 different random seeds, here we reduced the count due to computational cost to 4 for ZINC and CIFAR10, and 3 for PCQM4Mv2-Subset and MalNet-Tiny. All time measurements reported in this section are obtained on a system with identical hardware configuration: 1x NVidia A100 (40GB) GPU and allocation of 4 AMD Milan 7413 (2.65GHz) CPU cores.

Table B.1: GPS ablation study on **ZINC** dataset.

| GPS-MPNN | GPS-GlobAttn | PE / SE type | Test MAE ↓ | # Param. | Epoch / Total |
|----------|--------------|--------------|------------|----------|---------------|
| GINE | Transformer | RWSE-20 | $0.070 \pm 0.002$ | 423,717 | 14s / 7.56h |
| GINE | – | RWSE-20 | $0.070 \pm 0.004$ | 257,317 | 7s / 3.90h |
| GINE | Performer | RWSE-20 | $0.071 \pm 0.002$ | 913,317 | 18s / 9.85h |
| GINE | BigBird | RWSE-20 | $0.071 \pm 0.002$ | 507,557 | 38s / 21.20h |
| – | Transformer | RWSE-20 | $0.217 \pm 0.008$ | 340,517 | 10s / 5.74h |
| GatedGCN | Transformer | RWSE-20 | $0.086 \pm 0.002$ | 551,077 | 18s / 9.86h |
| PNA | Transformer | RWSE-20 | $0.070 \pm 0.003$ | 680,805 | 17s / 9.46h |
| GINE | Transformer | – | $0.113 \pm 0.007$ | 423,873 | 15s / 8.38h |
| GINE | Transformer | LapPE-8 | $0.116 \pm 0.009$ | 423,833 | 13s / 7.40h |
| GINE | Transformer | SignNet$^{\text{MLP}}$-8 | $0.090 \pm 0.007$ | 486,957 | 21s / 11.61h |
| GINE | Transformer | SignNet$^{\text{DeepSets}}$-37 | $0.079 \pm 0.006$ | 497,933 | 21s / 11.49h |
| GINE | Transformer | PEG$^{\text{LapEig}}$-8 | $0.936 \pm 0.143$ | 426,379 | 16s / 8.83h |
| GatedGCN | Transformer | PEG$^{\text{LapEig}}$-8 | $0.161 \pm 0.006$ | 553,739 | 20s / 11.07h |

Table B.2: Ablation study on **10% subset of PCQM4Mv2** with GPS-small (Appendix A).

| GPS-MPNN | GPS-GlobAttn | PE / SE type | Test MAE ↓ | # Param. | Epoch / Total |
|----------|--------------|--------------|------------|----------|---------------|
| GatedGCN | Transformer | RWSE-16 | $0.1159 \pm 0.0004$ | 6,152,001 | 61s / 1.70h |
| GatedGCN | – | RWSE-16 | $0.1213 \pm 0.0002$ | 4,297,601 | 45s / 1.26h |
| GatedGCN | Performer | RWSE-16 | $0.1142 \pm 0.0005$ | 5,855,601 | 83s / 2.30h |
| GatedGCN | BigBird | RWSE-16 | $0.1237 \pm 0.0022$ | 7,080,721 | 137s / 3.81h |
| – | Transformer | RWSE-16 | $0.3294 \pm 0.0137$ | 3,827,921 | 42s / 1.16h |
| GINE | Transformer | RWSE-16 | $0.1284 \pm 0.0037$ | 4,755,121 | 50s / 1.40h |
| PNA | Transformer | RWSE-16 | $0.1409 \pm 0.0131$ | 7,551,217 | 61s / 1.68h |
| GatedGCN | Transformer | – | $0.1355 \pm 0.0035$ | 6,155,089 | 59s / 1.63h |
| GatedGCN | Transformer | LapPE-8 | $0.1201 \pm 0.0003$ | 6,153,889 | 63s / 1.76h |
| GatedGCN | Transformer | SignNet$^{\text{MLP}}$-8 | $0.1158 \pm 0.0008$ | 6,217,013 | 87s / 2.41h |
| GatedGCN | Transformer | SignNet$^{\text{DeepSets}}$-21 | $0.1144 \pm 0.0002$ | 6,225,845 | 146s / 4.05h |
| GatedGCN | Transformer | PEG$^{\text{LapEig}}$-8 | $0.1209 \pm 0.0003$ | 6,162,390 | 67s / 1.86h |

Table B.3: Ablation study on **MalNet-Tiny**. *Configuration required decreased batch size.

| GPS-MPNN | GPS-GlobAttn | PE / SE type | Accuracy ↑ | # Param. | Epoch / Total |
|---|---|---|---|---|---|
| GatedGCN | Performer | – | 92.64 ± 0.78 | 527,237 | 46s / 1.90h |
| GatedGCN | – | – | 92.23 ± 0.65 | 199,237 | 6s / 0.25h |
| GatedGCN | *Transformer | – | 93.50 ± 0.41 | 282,437 | 94s / 3.94h |
| GatedGCN | BigBird | – | 92.34 ± 0.34 | 324,357 | 130s / 5.43h |
| – | Performer | – | 73.90 ± 0.58 | 421,957 | 41s / 1.73h |
| GINE | Performer | – | 92.27 ± 0.60 | 463,557 | 46s / 1.92h |
| PNA | Performer | – | 91.67 ± 0.70 | 592,149 | 47s / 1.97h |
| GatedGCN | Performer | LapPE-10 | 92.74 ± 0.45 | 527,701 | 47s / 1.91h |
| GatedGCN | Performer | RWSE-16 | 92.77 ± 0.31 | 527,425 | 46s / 1.90h |
| GatedGCN | Performer | SignNet$^{\text{MLP}}$-10 | 92.57 ± 0.40 | 591,063 | 65s / 2.72h |
| GatedGCN | Performer | *SignNet$^{\text{DeepSets}}$-32 | 93.13 ± 0.68 | 602,085 | 145s / 6.06h |
| GatedGCN | Performer | PEG$^{\text{LapEig}}$-10 | 92.27 ± 0.29 | 528,842 | 48s / 1.98h |

Table B.4: Ablation study on **CIFAR10**.

| GPS-MPNN | GPS-GlobAttn | PE / SE type | Accuracy ↑ | # Param. | Epoch / Total |
|---|---|---|---|---|---|
| GatedGCN | Transformer | LapPE-8 | 72.305 ± 0.344 | 112,726 | 62s / 1.72h |
| GatedGCN | – | LapPE-8 | 69.948 ± 0.499 | 79,654 | 43s / 1.18h |
| GatedGCN | Performer | LapPE-8 | 70.670 ± 0.338 | 239,554 | 77s / 2.14h |
| GatedGCN | BigBird | LapPE-8 | 70.480 ± 0.106 | 129,418 | 145s / 4h |
| – | Transformer | LapPE-8 | 68.862 ± 1.138 | 70,762 | 40s / 1.11h |
| GINE | Transformer | LapPE-8 | 71.105 ± 0.655 | 87,298 | 51s / 1.42h |
| PNA | Transformer | LapPE-8 | 73.418 ± 0.165 | 138,706 | 59s / 1.65h |
| GatedGCN | Transformer | – | 71.488 ± 0.187 | 112,590 | 61s / 1.69h |
| GatedGCN | Transformer | RWSE-16 | 71.958 ± 0.398 | 112,798 | 61s / 1.69h |
| GatedGCN | Transformer | SignNet$^{\text{MLP}}$-8 | 71.740 ± 0.569 | 175,850 | 116s / 3.21h |
| GatedGCN | Transformer | SignNet$^{\text{DeepSets}}$-16 | 72.368 ± 0.340 | 186,558 | 148s / 4.12h |
| GatedGCN | Transformer | PEG$^{\text{LapEig}}$-8 | 72.100 ± 0.460 | 113,529 | 67s / 1.87h |

Table B.5: Ablation study on **PascalVOC-SP** of LRGB [17]. Shown is the mean ± s.d. of 4 runs.

| GPS-MPNN | GPS-GlobAttn | PE / SE type | F1 ↑ | # Param. | Epoch / Total |
|---|---|---|---|---|---|
| GatedGCN | Transformer | LapPE-10 | 0.3736 ± 0.0158 | 510,453 | 17s / 1.46h |
| GatedGCN | – | LapPE-10 | 0.3016 ± 0.0031 | 361,461 | 8s / 0.68h |
| GatedGCN | Performer | LapPE-10 | 0.3724 ± 0.0131 | 1,148,277 | 25s / 2.09h |
| GatedGCN | BigBird | LapPE-10 | 0.2762 ± 0.0069 | 585,333 | 42s / 3.46h |
| – | Transformer | LapPE-10 | 0.2762 ± 0.0111 | 322,677 | 12s / 1.04h |
| GINE | Transformer | LapPE-10 | 0.3160 ± 0.0024 | 397,173 | 14s / 1.18h |
| PNA | Transformer | LapPE-10 | 0.3677 ± 0.0108 | 625,029 | 18s / 1.49h |
| GatedGCN | Transformer | – | 0.3846 ± 0.0156 | 510,069 | 17s / 1.4h |
| GatedGCN | Transformer | RWSE-16 | 0.3659 ± 0.0031 | 510,133 | 17s / 1.45h |
| GatedGCN | Transformer | SignNet$^{\text{MLP}}$-10 | 0.3473 ± 0.0051 | 573,869 | 41s / 3.4h |
| GatedGCN | Transformer | SignNet$^{\text{DeepSets}}$-48 | 0.3668 ± 0.0080 | 583,893 | 50s / 2.8h |
| GatedGCN | Transformer | PEG$^{\text{LapEig}}$-10 | **0.3956 ± 0.0084** | 512,281 | 19s / 1.6h |

Table B.6: Ablation study on **Peptides-func** of LRGB [17]. Shown is the mean ± s.d. of 4 runs.

| GPS-MPNN | GPS-GlobAttn | PE / SE type | AP ↑ | # Param. | Epoch / Total |
|---|---|---|---|---|---|
| GatedGCN | Transformer | LapPE-10 | 0.6535 ± 0.0041 | 504,362 | 6s / 0.35h |
| GatedGCN | – | LapPE-10 | 0.6159 ± 0.0048 | 355,370 | 3s / 0.16h |
| GatedGCN | Performer | LapPE-10 | 0.6475 ± 0.0056 | 748,970 | 11s / 0.61h |
| GatedGCN | BigBird | LapPE-10 | 0.5854 ± 0.0079 | 579,242 | 18s / 1.00h |
| – | Transformer | LapPE-10 | 0.6333 ± 0.0040 | 316,586 | 5s / 0.29h |
| GINE | Transformer | LapPE-10 | 0.6464 ± 0.0077 | 391,082 | 6s / 0.31h |
| PNA | Transformer | LapPE-10 | 0.6560 ± 0.0058 | 618,138 | 6s / 0.35h |
| GatedGCN | Transformer | – | 0.6214 ± 0.0326 | 506,506 | 6s / 0.33h |
| GatedGCN | Transformer | RWSE-16 | 0.6486 ± 0.0071 | 503,418 | 6s / 0.35h |
| GatedGCN | Transformer | SignNet$^{\text{MLP}}$-10 | 0.5840 ± 0.0140 | 568,726 | 41s / 3.39h |
| GatedGCN | Transformer | SignNet$^{\text{DeepSets}}$-48 | 0.6314 ± 0.0059 | 577,802 | 49s / 2.73h |
| GatedGCN | Transformer | PEG$^{\text{LapEig}}$-10 | 0.6461 ± 0.0047 | 508,718 | 19s / 1.60h |

# C Theoretical results

## C.1 Why do we need PE and SE?

In this section, we review the 1-Weisfeiler-Leman test [59], their equivalence with MPNNs and the limitations brought by this equivalent expressive power which eventually brings us to a statement that indicates the theoretical need of equipping MPNNs or GTs with either or a combination of local, relative or global PE/SE.

**1-Weisfeiler-Leman test (1-WL).** The 1-WL test is a node-coloring algorithm, in the hierarchy of Weisfeiler-Leman (WL) heuristics for graph isomorphism, [59], which iteratively updates the color of a node based on its 1-hop local neighborhood until an iteration when the node colors do not change successively. The final histogram of the node colors determine whether the algorithm outputs the two graphs to be 'non-isomorphic' (when the histograms of 2 graphs are distinct) or 'possibly isomorphic' (when the histograms of 2 graphs are same). Although, it is not a sufficient test for the graph isomorphism problem, the heuristic is simple to apply and has been popularly used in the literature recently to quantify the expressive power of MPNNs.

**Expressive power of MPNNs.** Based on the equivalence of the aggregate and update functions of MPNNs with the hash function of the 1-WL test, it was shown that MPNNs are at most powerful as 1-WL [61, 45], which is now popularly understood in the literature. Graph Isomorphism Network [61] was developed by aligning the injectivity of the aggregate and update functions of GIN with the injectivity of the 1-WL's hash function, which makes it a 1-WL powerful MPNN. In direct consequence, the power of the GIN is quantified as 1-WL expressive, *i.e.*, if 1-WL outputs two graphs to be 'non-isomorphic' then the GIN would output different feature vectors for the two graphs and conversely, if 1-WL outputs two graphs to be 'possibly isomorphic', the feature embeddings of the two graphs would be the same. We refer the readers to [61] for the details on this theoretical result.

Since the expressive power of MPNNs are at most 1-WL, it leads to a serious limitation in distinguishing a wide-variety of non-isomorphic graphs [50]. Note that numerous follow up works have proposed GNNs that are strictly powerful than 1-WL, often moving away from the message passing framework [22] on which MPNNs are based [45, 10, 43]. As higher-order GNNs are not within the scope of this section, we limit our discussion only to MPNNs, such as GINs, which makes them 1-WL powerful. There are numerous examples on which MPNNs fail as a result [50]. Among such cases, we consider two examples tasks: the task to differentiate between two non-isomorphic Circular Skip Link (CSL) graphs, Figure C.1a, and the task to differentiate between two potential links, Figure C.1b. The nodes in these examples do not have discriminating node features.

**The CSL graph, Figure C.1a.** In the CSL graph-pair [46], the two graphs $\mathcal{G}_{\text{skip}}(11, 2)$ and $\mathcal{G}_{\text{skip}}(11, 3)$ differ in the length of skip-link of a node and are hence non-isomorphic. Since the 1-WL algorithm produces the same color for all the nodes in both graphs, MPNNs will generate similar node colors. See the colors generated by 1-WL and MPNN in the second row of Figure C.1a. However, the use of a *global* PE (eg. Laplacian PE [15]) assigns each node a unique color, as depicted in the third row. Consequently, the feature embeddings of the two graphs which are the hash function outputs of the collection of node colors are different, thus making the task to distinguish the graphs successful. Similarly, the use of a *local* SE (e.g. diagonals of $m$-steps random walk) allows the coloring of the nodes of the 2 graphs to be different [16] since it captures the difference of the skip links of the two graphs successfully [42]. See the fourth row where the local SE based colors are depicted on the nodes. Therefore, either of the specific *local* SE or *global* PE can help distinguish the two graphs which cannot be learnt by 1-WL or MPNNs.

**The Decalin molecular graph, Figure C.1b.** In the Decalin graph, the node $a$ is isomorphic to node $b$, and so is the node $c$ to node $d$. A 1-WL coloring of the nodes, and equivalently MPNN, would generate one color for the node $a, b$ and another color for $c, d$, see the second row in Figure C.1b. If that task is to identify a potential link between the node-sets $(a, d)$ and $(b, d)$, the combination of the node colors of the node-sets will produce the same embedding for the two links, thus making the 1-WL or MPNNs based coloring unsuitable to certain tasks [67]. A similar observation also follows for the node coloring based on the aforementioned *local* SE [16], which is illustrated in the fourth row in Figure C.1b. However, using a distance-based *relative* PE on the edges or an eigenvector-based

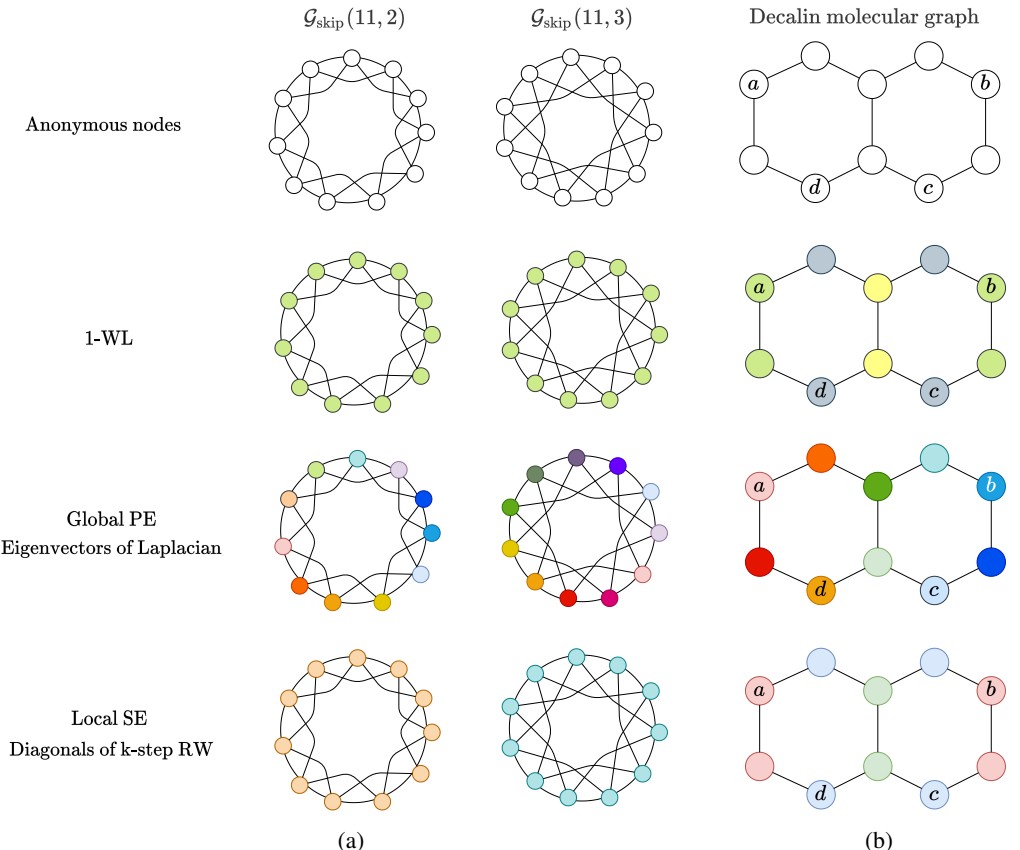

Figure C.1: **First Row:** Example graphs with anonymous nodes, *i.e.*, nodes do not have any distinguishing node features. (a) A pair of Circular Skip Link (CSL) graphs [46] where the nodes have skip links of 2 and 3 respectively. (b) A Decalin molecular graph which has two rings of all Carbon atoms, thus with no distinguishing node features. **Second Row:** The nodes colored with the feature generated by 1-WL [59, 61, 45]. **Third Row:** The nodes colored with the feature generated by *global* PE [15]. **Fourth Row:** The nodes colored with the feature generated by *local* SE [16]. *Note:* The colors depicted on nodes in the graphs represent a unique feature vector generated, for a given graph, from the corresponding PE/SE. Figure best visualized in color.

*global* PE would successfully differentiate the embeddings of the two links. Therefore, the *relative* PE or the *global* PE which can help to distinguish between the two links cannot be learnt by 1-WL or MPNNs.

We can then conclude the following statement based on the above discussion which provides a theoretical basis for the need of PE and SE, as the PE and SE can be directly supplying essential information for the task:

**Proposition 1.** *Assuming no modification applied to MPNNs for a learning task, there exists Positional Encodings (PE) and Structural Encoding (SE) which MPNNs are not guaranteed to learn.*

## C.2 Preserving edge information in the self-attention layer

In this section, we argue that an MPNN layer is able to propagate the information from edges to nodes such that, when computing the attention between nodes, the global Attention (Transformer) layer can infer whether two nodes are connected and what are the edge features between them.

Suppose an MPNN with the sum aggregator, with the update function as given below:

$$h_u^{l+1} = \sum_{v \in \mathcal{N}_u} f(h_u^l, h_v^l, e_{uv}), \tag{5}$$

where $f$ is a learned function, e.g., an MLP; $u$ is the index of a central node whose neighborhood is being aggregated; $v$ is the index of a neighbor of $u$; $h_u^l$ the node features at layer $l$ for node $u$, and $e_{uv}$ the edge features between nodes $u$ and $v$.

We know from the Lemma 5 of Xu et al. [61] that the sum over a countable multiset is universal, meaning it can map a unique multiset to any possible function. Let's assume that $h_u$ is unique and countable for every node $u$, which can be accomplised using all the Laplacian eigenvectors as PE. Then, there exist a function $f$ such that an encoding $\mu_{uv}$ that respects the following characteristics is propagated to the nodes: (i) unique for the triplet $\{h_u, h_v, e_{uv}\}$, (ii) invariant to the permutation of $u$ and $v$, (iii) contains the information of $e_{ij}$, (iv) all information of $\mu_{uv}$ is preserved after the $\sum$.

Hence, an Attention layer that follows the message-passing is able to infer whether two nodes are connected since both nodes will contain the unique identifier $\mu_{uv}$, and will also be able to infer the edge features from it.

An example of such function $\mu_{uv}$ is the tensor product $\otimes$ of a one-hot encoding unique for each edge $o_{uv}$ and the edge features $e_{uv}$. For example, if $e_{uv} = [e_1, e_2, e_3]$ and the edge is represented with $o_{uv} = [0, 1, 0, 0]$, then $\mu_{uv} = o_{uv} \otimes e_{uv} = [0, 0, 0, e_1, e_2, e_3, 0, 0, 0, 0, 0, 0]$ satisfies all the above conditions. Although this function requires an exponential increase in the hidden dimension, this is also the case for the Lemma 5 in Xu et al. [61].

# D  GPS schematics

## D.1  GPS layer

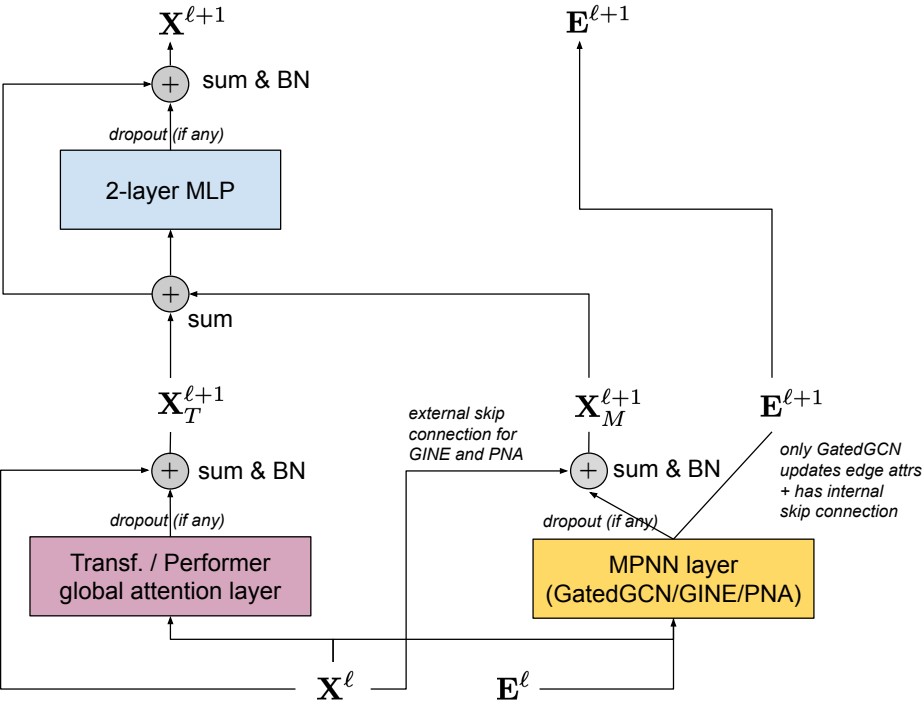

Figure D.1: Modular GPS layer that combines local MPNN and global attention blocks. Local MPNN encodes real edge features into the node-level hidden representations, while global attention mechanism can implicitly make use of this information together with PE/SE to infer relation between two nodes without explicit edge features. After each functional block (an MPNN layer, a global attention layer, an MLP) we apply residual connections followed by batch normalization (BN) [30]. In the 2-layer MLP block we use ReLU activations and its inner hidden dimension is twice the layer-input feature dimensionality $d_\ell$. Note, similarly to Transformer, the input and output dimensionality of the GPS-layer as a whole is the same.

**GPS layer equations.** In Section 3.3 of the main text we provide a simplify definition of the GPS computational layer for clarity, here we additionally list the precise application of skip connections, dropout, and batch normalization with learnable affine parameters:

$$\mathbf{X}^{\ell+1}, \mathbf{E}^{\ell+1} = \texttt{GPS}^\ell \left( \mathbf{X}^\ell, \mathbf{E}^\ell, \mathbf{A} \right) \tag{6}$$

$$\text{computed as} \quad \hat{\mathbf{X}}_M^{\ell+1}, \mathbf{E}^{\ell+1} = \texttt{MPNN}_e^\ell \left( \mathbf{X}^\ell, \mathbf{E}^\ell, \mathbf{A} \right), \tag{7}$$

$$\hat{\mathbf{X}}_T^{\ell+1} = \texttt{GlobalAttn}^\ell \left( \mathbf{X}^\ell \right), \tag{8}$$

$$\mathbf{X}_M^{\ell+1} = \texttt{BatchNorm} \left( \texttt{Dropout} \left( \hat{\mathbf{X}}_M^{\ell+1} \right) + \mathbf{X}^\ell \right), \tag{9}$$

$$\mathbf{X}_T^{\ell+1} = \texttt{BatchNorm} \left( \texttt{Dropout} \left( \hat{\mathbf{X}}_T^{\ell+1} \right) + \mathbf{X}^\ell \right), \tag{10}$$

$$\mathbf{X}^{\ell+1} = \texttt{MLP}^\ell \left( \mathbf{X}_M^{\ell+1} + \mathbf{X}_T^{\ell+1} \right) \tag{11}$$

## D.2 GPS algorithm

---

**Algorithm 1** Algorithm for an $L$ layer GPS network.

---

**Input:** Graph $\mathcal{G} = (\mathcal{V}, \mathcal{E})$ with $N$ nodes and $E$ edges; Adjacency matrix $\mathbf{A} \in \mathbb{R}^{N \times N}$; Node features $\mathbf{X} \in \mathbb{R}^{N \times D_{\text{node}}}$; Edge features $\mathbf{E} \in \mathbb{R}^{E \times D_{\text{edge}}}$; Local message passing model instance $\texttt{MPNN}_e$; Global attention model instance $\texttt{GlobalAttn}$; Positional Encoding function $F_{\text{PE}}$; Structural Encoding function $F_{\text{SE}}$; Layer $\ell \in [0, L-1]$.
**Output:** Node representations $\mathbf{X}^L \in \mathbb{R}^{N \times D}$ and edge representations $\mathbf{E}^L \in \mathbb{R}^{E \times D}$, that can downstream be composed with appropriate *prediction head* for graph, node, or edge -level prediction.

1. $\mathbf{P}_{\text{node}}, \mathbf{P}_{\text{edge}}, \mathbf{S}_{\text{node}}, \mathbf{S}_{\text{edge}} \leftarrow \emptyset$

2. if $F_{\text{PE}}$ is relative then $\mathbf{P}_{\text{edge}} \leftarrow F_{\text{PE}}(\mathcal{G}) \in \mathbb{R}^{E \times D_{\text{PE}}}$ else $\mathbf{P}_{\text{node}} \leftarrow F_{\text{PE}}(\mathcal{G}) \in \mathbb{R}^{N \times D_{\text{PE}}}$

3. if $F_{\text{SE}}$ is relative then $\mathbf{S}_{\text{edge}} \leftarrow F_{\text{SE}}(\mathcal{G}) \in \mathbb{R}^{E \times D_{\text{SE}}}$ else $\mathbf{S}_{\text{node}} \leftarrow F_{\text{SE}}(\mathcal{G}) \in \mathbb{R}^{N \times D_{\text{SE}}}$

4. $\mathbf{X}^0 \leftarrow \bigoplus_{\text{node}} \left( \texttt{NodeEncoder}(\mathbf{X}), \mathbf{P}_{\text{node}}, \mathbf{S}_{\text{node}} \right) \in \mathbb{R}^{N \times D}$

5. $\mathbf{E}^0 \leftarrow \bigoplus_{\text{edge}} \left( \texttt{EdgeEncoder}(\mathbf{E}), \mathbf{P}_{\text{edge}}, \mathbf{S}_{\text{edge}} \right) \in \mathbb{R}^{E \times D}$

6. for $\ell = 0, 1, \cdots, L-1$

   (a) $\hat{\mathbf{X}}_M^{\ell+1}, \mathbf{E}^{\ell+1} \leftarrow \texttt{MPNN}_e^{\ell} \left( \mathbf{X}^\ell, \mathbf{E}^\ell, \mathbf{A} \right)$

   (b) $\hat{\mathbf{X}}_T^{\ell+1} \leftarrow \texttt{GlobalAttn}^{\ell} \left( \mathbf{X}^\ell \right)$

   (c) $\mathbf{X}_M^{\ell+1} \leftarrow \texttt{BatchNorm} \left( \texttt{Dropout} \left( \hat{\mathbf{X}}_M^{\ell+1} \right) + \mathbf{X}^\ell \right)$

   (d) $\mathbf{X}_T^{\ell+1} \leftarrow \texttt{BatchNorm} \left( \texttt{Dropout} \left( \hat{\mathbf{X}}_T^{\ell+1} \right) + \mathbf{X}^\ell \right)$

   (e) $\mathbf{X}^{\ell+1} \leftarrow \texttt{MLP}^{\ell} \left( \mathbf{X}_M^{\ell+1} + \mathbf{X}_T^{\ell+1} \right)$

7. return $\mathbf{X}^L \in \mathbb{R}^{N \times D}$ and $\mathbf{E}^L \in \mathbb{R}^{E \times D}$

---

where $\bigoplus$ denotes an operator for combining the input node or edge features with their respective positional and/or structural encoding, in practice this is a concatenation operator which can be changed to sum or other operators; $\texttt{NodeEncoder}$ and $\texttt{EdgeEncoder}$ are dataset-specific initial node and edge feature encoders potentially with learnable parameters; $\texttt{MPNN}_e$ and $\texttt{GlobalAttn}$ have their corresponding learnable parameters at each layer $\ell$; $\hat{\mathbf{X}}_M^{\ell+1}$ and $\hat{\mathbf{X}}_T^{\ell+1}$ denote the intermediate node representations given by the local message passing module and the global attention module respectively; and $\texttt{MLP}^{\ell}$ is a multi layer perceptron module with its own learnable parameters that combines the intermediate $\mathbf{X}_M^{\ell+1}$ and $\mathbf{X}_T^{\ell+1}$. Note that a relative $F_{\text{PE}}$ or $F_{\text{SE}}$ produces PE or SE for each edge which are thence handled accordingly in lines 2 and 3 in Algorithm 1.