# OpenReview forum: "Recipe for a General, Powerful, Scalable Graph Transformer"
_NeurIPS.cc/2022/Conference — NeurIPS 2022 Accept_

### Official Review · Reviewer_TX95 · 2022-07-11

**Rating:** 5
**Confidence:** 4
**Soundness:** 2 fair
**Presentation:** 3 good
**Contribution:** 2 fair

**Summary:**

This paper provides a general, powerful, scalable (GPS) graph Transformer with linear complexity. The authors redefine positional encodings (PEs) and structural encodings (SEs) with local, global, and relative categories and try to incorporate PEs and SEs with local and global attention in a graph Transformer. With the proposed GPS layers, the authors show the competitive result on several datasets. The proposed definition of PE and SE is general, and the paper has a good presentation of writing logic. However, I have several problems about the methods and experiments.

**Questions:**

refer to "limitations"

**Limitations:**

1. The PE and SE categories of local, global and relative are confusing. The local and global are opposite, but relative is another dimension, and it should be more suitable compared with the absolute PE.
2. The description of relative SE is inconsistent with its example. The authors claimed that description allows two nodes to understand how much their structures differ, but examples are not enough to illustrate. Relative SE lacks support from related work, and this paper has no novelty on this point. Relative SE is more like a definition forcibly created to correspond to relative PE.
3. The contribution of this paper is unclear. The first point should be merged with the third point, and the fifth point should not be regarded as the contribution of this paper.
4. All methods in the paper are summary and induction, which lack novelty. There is no key innovation part of the paper.
5. The result of SAN on ogbg-molhiv dataset misses the decimal point and zero in Table 4.

**Strengths And Weaknesses:**

1. The authors summarize the previous works about positional encodings and propose novel categories of structural encodings.
2. The authors explain the rationality of definition and categories by the 1-Weisfeiler-Leman test and Circular Skip Link (CSL) graph.
3. Based on PEs and SEs, the authors propose the GPS layer and present its characteristics with theoretical analysis.
4. The paper has good writing and logical structure.

---

> ### Author Response · Authors · 2022-08-02
> **Response to Reviewer TX95 (2/2)**
>
> ### Re Q4:
> We believe our work provides a set of contributions to the graph learning community worthy of NeurIPS; summarized in the 5 points at the end of the Introduction section. Our approach is well motivated and the fact that the 3-part recipe turns out to be relatively straightforward is only to its benefit. We evaluated GPS on the largest set of benchmarking datasets of any single graph Transformer paper (outperforming every graph Transformer on 10 out of 11 benchmarks), and provided crucial insights in ablation studies.
>
> ### Re Q5:
> Thank you for catching this formatting issue, it is fixed in the revised version!
>
>
> [2] Kreuzer, D. et al. "Rethinking graph transformers with spectral attention." NeurIPS 2021
>
> [6] Dwivedi, V.P., and Bresson X. "A generalization of transformer networks to graphs." arXiv:2012.09699 (2020).
>
> [7] Bodnar et al. “Weisfeiler and Lehman Go Cellular: CW Networks”. NeurIPS 2021

---

> ### Author Response · Authors · 2022-08-02
> **Response to Reviewer TX95 (1/2)**
>
> We thank the reviewer for their review and are happy to hear that they found our paper well structured and well written. We carefully considered their criticism and hope we can convince the reviewer that the paper is worth a higher score by clearing up their concerns.
>
> ### Re Q1:
> The local and global PE/SE together are in a sense “absolute”, they differ in the scope of the frame in which they provide the position (local or global). Thinking of them as opposites may not be the best way. Perhaps we could rephrase our categorization as firstly differentiating absolute and relative encodings, and secondly further dividing the absolute encodings to local and global.
> With that, we note that our categorization is not definitive as some existing PE/SE do not necessarily fall exclusively under one category (as ​​mentioned in the caption of Table 1, page 4) . For example SignNet, which starts with global PE features (Laplacian eigenvectors) but then employs a learnable 8-layer GIN model that can contribute aspects of relative PE (by computing relative distance) and local SE (as GIN could learn to be e.g. the WL kernel). With more emerging graph positional and structural encodings, a different categorization or taxonomization may become needed. Yet we believe the categorization as we put it forward provides a useful frame of reference and axes along which to analyze graph PE/SE properties.
>
> ### Re Q2:
> Relative SE is indeed an uncommon category, it nevertheless exists. In Table 1 we list one such example based on a recently published method, the CW Networks by Bodnar et al. [7] that perform sub-structure aware message passing, equivalent to a 1/0 weighting of the edges belonging to the same/different sub-structures. Further, it is easy to imagine using relative node distances based on graph kernels (e.g., RW or k-WL kernels) as relative SEs. Thus it certainly has a place in our PE/SE categorization.
>
> ### Re Q3:
> We stand behind our 5 main contribution points that are clearly listed in the Introduction and summarized in the Conclusion. Here we expand on the contribution points (i), (iii), and (v):
>
> **(i) point:** Our main contribution is the general “GPS” blueprint which incorporates 3 principal blocks: positional and structural encodings, local message passing and global attention. To the best of our knowledge, such a blueprint has not been investigated in the existing literature. Although some related works have been done, which we discuss in Section 2. A key distinguishing aspect of our work is the clear separation of these 3 blocks and recognition of their principal inductive biases. The current literature typically focuses on one kind of PE/SE, there is no consensus around a universal PE/SE. Next, the differentiation of the local and global attention components, emphasizes the importance of the “locality bias” of MPNNs and the necessity for more efficient information propagation across a graph. This yields a design space that is considerably more flexible and applicable to a variety of graph learning datasets, as demonstrated by our extensive experimental results: outperforming every graph Transformer on 10 out of 11 benchmarks.
>
> **(iii) point:** All published graph Transformers utilize a quadratic attention mechanism. We are the first to accomplish linear (in the number of nodes) scalability of global attention-enabled GNN and demonstrate it on the MalNet-Tiny dataset with graphs of several thousands of nodes. While we achieve this in part thanks to the modularity of our design (the (i) contribution), we understand it as a separate contribution worthy of dedicated attention. In the response to Reviewer Md1W we further discuss the pros and cons of linear attention via Performer kernel approach.
>
> **(v) point:** We provide the GraphGPS package which is built on top of GraphGym – a design space for GNNs. In a similar vein, GraphGPS is a codebase for a design space of graph Transformers, with support for a variety of MPNNs, graph Transformers, and PE/SEs. We take this opportunity to emphasize that the package is not simply a code release of the paper for reproducibility purposes, which is implicit. But, beyond that, we reimplemented SAN [2] in GraphGPS as well as the original Graph Transformer [6], both of which are noticeably faster than their original implementations. As such, our fifth contribution is the open-source package that implements the very modular GPS blueprint, provides a testbed for new positional and structural encodings (irrespective of the actual GNN used, e.g. a standard MPNN without any global attention is perfectly well supported too), MPNNs and global attention mechanisms. We believe GraphGPS to be a convenient resource for researchers given its modular implementation and support for over a dozen of existing benchmarking datasets, which is easily extensible as well.

---

> > ### Comment · Reviewer_TX95 · 2022-08-03
> > **thanks for your clarification**
> >
> > after reading this reply, I think this work is helpful for the community of graph learning. Although its theoretical contribution is limited, I appreciate these take-aways for researchers and practitioners and really like the availability of this work through GraphGym. Thus, I just changed the overall score from reject to borderline accept.

---

### Official Review · Reviewer_Md1W · 2022-07-11

**Rating:** 7
**Confidence:** 4
**Soundness:** 4 excellent
**Presentation:** 3 good
**Contribution:** 3 good

**Summary:**

The authors present a way how to efficiently use transformers on graph data. They report modular architecture, containing of graph positional encodings, structural encodings, and graph features, that further passed to an ensemble of arbitrary transformer block and arbitrary message passing neural network.

**Questions:**

How the model performs if we use another architecture instead of transformer block? Say simple MLP?

**Ethics Review Area:**

["I don’t know"]

**Limitations:**

The authors addressed the potential societal impact and limitations in the collusion section.

**Strengths And Weaknesses:**

The paper deals with an important and trending task. It's a nice and interesting read overall, the paper is technically sound, and well structured and describes the problem well. The authors provided performance testing on sufficient amount of datasets, and show the performance deviation with different random seeds. The authors provided a package built on top of graphGym which is a big plus.

On the other hand, the solid proof to use transformer block is missing. The authors claim that their solution has linear complexity, however from the table 2 is clear that linear attention block gives only marginal performance improvement, when for standard tasks (like LRA benchmark) linear transformers are on par, or even outperform vanilla transformer.

---

> ### Author Response · Authors · 2022-08-02
> **Response to Reviewer Md1W**
>
> Our GPS empirically benefits from global attention, however we observed the magnitude of this performance gain to be dataset-dependent, with most pronounced benefit in datasets with long-range dependencies. Please see our answer to Question 1 of Reviewer **719v**.
>
> The pros and cons of linear attention mechanism vs. $O(N^2)$ vanilla Transformer attention is an interesting discussion point that we would like to expand on.
> - In the majority of current datasets we do not observe the performance benefits of linear attention. The average size of graphs in the majority of current benchmarking datasets does not surpass a few hundreds, which is manageable for an $O(N^2)$ Transformer. Here it is important to mention that in practice, even though the asymptotic complexity remains quadratic, our GPS design still empirically benefits from not having to explicitly condition the Transformer attention. This is in contrast to SAN or Graphormer, that need to explicitly construct the dense attention matrix conditioned on other graph properties, such as shortest-path-distances or edge types and attributes. As we mention in the main text, GPS with local MPNN and Transformer is in practice much faster than SAN despite the same asymptotic complexity. Therefore our design choice of decoupling local processing from the global attention not just allows for a plug-and-play linear attention (or many other x-former models), it also significantly improves wall-clock run time when $O(N^2)$ Transformer is used.
> - Performer (linear attention) starts to provide meaningful speedup once the graphs are several thousands of nodes large. We demonstrated this point in our ablation studies on MalNet-Tiny, where GPS with a Performer global attention module is approximately twice as fast as with a vanilla Transformer (Table B.3).
> - In practice, we observed that as long as the $O(N^2)$ Transformer is not prohibitively expensive (i.e., it would be graphs with above ~10k nodes) it remains the best choice for a global attention module. The Transformer tends to yield better prediction performance than Performer in all our experiments. This is in line with the recent findings in NLP applications of Transformer-like models by Tay et al. [4]. Linear transformers (or x-formers in general) have a different inductive bias and scale differently at varying dataset size, model size, and compute budget scenarios, often leaving “vanilla” Transformer as the best choice [4,5].
>
> We did not include this discussion in the current revision of our paper. Upon acceptance, we would include it in a potential camera ready version that allows an additional space to accommodate additional content.
>
>
> [4] Tay, Yi, et al. "Scaling Laws vs Model Architectures: How does Inductive Bias Influence Scaling?." arXiv:2207.10551 (2022).
>
> [5] Dehghani, M. et al. "The efficiency misnomer." ICLR 2022, arXiv:2110.12894 (2022).​​

---

> > ### Comment · Reviewer_Md1W · 2022-08-08
> > **Thank you for your response.**
> >
> > Thank you for your clarification on the subquadratic transformers.

---

### Official Review · Reviewer_719v · 2022-07-12

**Rating:** 6
**Confidence:** 4
**Soundness:** 3 good
**Presentation:** 3 good
**Contribution:** 3 good

**Summary:**

In this work, the authors propose a recipe on how to build a general, powerful, scalable  graph Transformer with linear complexity and state-of-the-art results on a diverse set of benchmarks.

**Questions:**


*  Many recent studies show that the dense attention map is not necessary in the Transformer. In graph Transformer, Do we really need to allow nodes to attend to all other nodes in a graph (global attenion).
* The universal function on graph is not clear. A more detailed analysis why the proposed framework can achieve universal funtion approximator is required.

**Ethics Review Area:**

["I don’t know"]

**Limitations:**

Yes

**Strengths And Weaknesses:**

Strengths:

1. The proposed model is scalable due to its linear comlexity.

2. The work considers lots of message:  positional and structural encodings with local message passing and global attention.

3. The code is available and the performance of this work is good.

Weaknesses:
1. The overall algorithm can be well presented, like a presudo algorithm can be helpful to understand the implementation.

2.   Many recent studies show that the dense attention map is not necessary in the Transformer. In graph Transformer, Do we really need to allow nodes to attend to all other nodes in a graph (global attenion).

3. The universal function on graph is not clear. A more detailed analysis why the proposed framework can achieve universal funtion approximator is required.

---

> ### Author Response · Authors · 2022-08-02
> **Response to Reviewer 719v**
>
> We would like to thank the Reviewer for the valuable feedback and comments on the limitations and questions.
>
> ### W1: The overall algorithm can be well presented, like a presudo algorithm can be helpful to understand the implementation.
>
> Please see Appendix D for exact formulation of the GPS layer and Figure 1 for the flowchart of the whole GPS pipeline. Additionally, in the current revision, we extended the Appendix D with a GPS pseudocode algorithm, please see the new Appendix D.2 (page 24).
>
> ### W2 / Q1: In graph Transformer, Do we really need to allow nodes to attend to all other nodes in a graph (global attention).
>
> We asked ourselves the same question, i.e. whether and when is global attention beneficial. We investigated it in two ways. 1) We conducted ablation studies, where we disabled the global attention module in GPS. These results are part of the original submission, Table 2A, with all the details in Appendix B. Indeed, the global attention is not always necessary, and its usefulness is dataset-dependant. 2) We suspect that global attention is particularly important in datasets that contain long-range dependencies. Therefore we utilized a recently proposed Long-Range Graph Benchmark (LRGB) [1] set of 5 such datasets. These results are part of our revised paper, see Appendix E. In all these benchmarks, GPS with global attention outperformed MPNN baselines by a large margin. Further, additional ablation studies (Tables E.2 and E.3) confirm that disabling of the global-attention module in GPS leads to notable performance degradation on these datasets with long-range dependencies.
>
> ### W3 / Q2: The universal function on graph is not clear. A more detailed analysis why the proposed framework can achieve universal funtion approximator is required.
>
> Although we mostly refer to the theoretical result of Kreuzer et al. for SAN [2] to justify the universality of GPS in the main text, we also provide more details about theoretical expressiveness in Appendix C.
>
> In particular, the seminal work of Xu et al. [3] showed that under the assumption of exponential increase of hidden dimension, the sum over a countable multiset is universal. It is also known [3] that if all node representations $h_u$ are unique and countable for every node $u$, then there exists an injective permutation-invariant function as long as this hashing includes *all* information about the edges. The intuition behind universality of the GraphGPS architecture lies in that: (1) the uniqueness of $h_u, \forall u \in G$ is achieved via structural and/or positional encodings, e.g., using the Laplacian eigenvectors as PEs that can be aggregated by any set function like DeepSets and added to node features. And (2), the unique hashing is achieved by the self-attention mechanism over an expressive enough function that can be, for instance, a tensor product of one-hot encoding unique for each edge with its edge feature (Appendix C.2). Such a function requires an exponential increase of the node representation size with each added layer, but so does the original proof by Xu et al [3], so we are not further relaxing their assumptions.
>
> We agree that the argumentation in the original paper revision might feel superficial, but we commit to improve that in the final version, adding an extended section based on the explanation provided herein.
>
>
> [1] Anonymous et al. Long range graph benchmark. Under review in NeurIPS Dataset and Benchmarking track, 2022. [Note: the PDF is included in the revised Supplementary Material ZIP file.]
>
> [2] Kreuzer, D. et al. "Rethinking graph transformers with spectral attention." NeurIPS 2021
>
> [3] Xu et al. How Powerful are Graph Neural Networks? ICLR 2019

---

> > ### Author Response · Authors · 2022-08-08
> > **Inviting your feedback**
> >
> > Dear Reviewer 719v,
> >
> > Thank you again for your review and comments! We have tried our best to address your questions and accordingly we revised the paper.
> >
> > As we are near the end of the discussion period, we sincerely hope that you could provide us with a feedback on our revision and whether it has addressed your concerns. If so, we would appreciate it if you could consider raising the score. And if not, please let us know your outstanding concerns!
> >
> > Best regards,
> > Authors

---

### Author Response · Authors · 2022-08-02
**First Revision Summary**

We thank all reviewers for their time and reviews! We carefully considered the points they raised and we answer them in direct replies. Please note that we have uploaded a revised version of the paper. This revision has minimal changes to the main text, a few minor fixes/typos. However we extended the Appendix, which is now included in the revised main paper PDF (instead of a separate file in the supplementary ZIP):
- Added a new section, Appendix E, with additional results on Long range graph benchmark [1].
- Extended Appendix D by a GPS pseudocode algorithm, now in Appendix D.2.

[1] Anonymous et al. Long range graph benchmark. Under review in NeurIPS Dataset and Benchmarking track, 2022. [Note: the PDF is included in the revised Supplementary Material ZIP file.]

---

### Meta-Review · Area_Chair_dzUB · 2022-08-24

**Recommendation:** Accept
**Confidence:** Certain

**Metareview:**

This paper presents a powerful, general, scalable, and linearly complex graph Transformer. Positional encodings and structural encodings are redefined with local, global, and relative categories, and an attempt has been made to include  local and global focus attentions in a graph Transformer. All of the reviewers acknowledged the novelty of this work, particularly within the context of the domain, and therefore voted for its acceptance. Please take feedback from reviewers into account when preparing the camera-ready version.

**Award:**

No

---

### Decision · Program_Chairs · 2022-09-14

Accept